# Base editing strategies to convert CAG to CAA diminish the disease-causing mutation in Huntington's disease

**Doo Eun Choi[1,2], Jun Wan Shin[1,2], Sophia Zeng[1], Eun Pyo Hong[1,2,3], Jae-Hyun Jang[1,2], Jacob M Loupe[1,2], Vanessa C Wheeler[1,2], Hannah E Stutzman[1,4], Ben Kleinstiver[1,4,5], Jong-Min Lee[1,2,3]\***

[1]Center for Genomic Medicine, Massachusetts General Hospital, Boston, United States; [2]Department of Neurology, Harvard Medical School, Boston, United States; [3]Medical and Population Genetics Program, The Broad Institute of MIT and Harvard, Cambridge, United States; [4]Department of Pathology, Massachusetts General Hospital, Boston, United States; [5]Department of Pathology, Harvard Medical School, Boston, United States

## eLife Assessment

This proof-of-concept study focuses on an A->G DNA base editing strategy that converts CAG repeats to CAA repeats in the human HTT gene, which causes Huntington's disease (HD). These studies are conducted in human HEK293 cells engineered with a 51 CAG canonical repeat and in HD knock-in mice harboring 105+ CAG repeats. The findings of this study are **valuable** for the HD field, applying state-of-the-art techniques; however, the key experiments have yet to be performed in neuronal systems or brains of these mice: actual disease-rectifying effects relevant to patients have yet to observed, leaving the work **incomplete**.

**\*For correspondence:**
JLEE51@mgh.harvard.edu

**Abstract** An expanded CAG repeat in the huntingtin gene (*HTT*) causes Huntington's disease (HD). Since the length of uninterrupted CAG repeat, not polyglutamine, determines the age-at-onset in HD, base editing strategies to convert CAG to CAA are anticipated to delay onset by shortening the uninterrupted CAG repeat. Here, we developed base editing strategies to convert CAG in the repeat to CAA and determined their molecular outcomes and effects on relevant disease phenotypes. Base editing strategies employing combinations of cytosine base editors and guide RNAs (gRNAs) efficiently converted CAG to CAA at various sites in the CAG repeat without generating significant indels, off-target edits, or transcriptome alterations, demonstrating their feasibility and specificity. Candidate BE strategies converted CAG to CAA on both expanded and non-expanded CAG repeats without altering *HTT* mRNA and protein levels. In addition, somatic CAG repeat expansion, which is the major disease driver in HD, was significantly decreased in the liver by a candidate BE strategy treatment in HD knock-in mice carrying canonical CAG repeats. Notably, CAG repeat expansion was abolished entirely in HD knock-in mice carrying CAA-interrupted repeats, supporting the therapeutic potential of CAG-to-CAA conversion strategies in HD and potentially other repeat expansion disorders.

## Introduction

Huntington's disease (HD; MIM #143100) (*Huntington, 1872*; *Macdonald, 1993*; *Bates et al., 2015*) is one of many trinucleotide repeat disorders caused by expansions of CAG repeats (*Gusella and*

*MacDonald, 2000*; *Ross, 2002*; *Di Prospero and Fischbeck, 2005*; *Orr and Zoghbi, 2007*; *Depienne and Mandel, 2021*). Although the underlying causative genes, pathogenic mechanisms, clinical features, and target tissues may be different (*Orr and Zoghbi, 2007*; *Paulson et al., 2000*; *Gatchel and Zoghbi, 2005*), these disorders share a cardinal feature: an inverse relationship between age-at-onset and respective CAG repeat length (*Gusella and MacDonald, 2000*; *Orr and Zoghbi, 2007*; *Orr et al., 1993*; *Pulst et al., 1996*; *Stevanin et al., 2000*; *Zoghbi and Orr, 2000*; *Schöls et al., 2004*; *Pearson et al., 2005*; *Andrew et al., 1993*; *Duyao et al., 1993*; *Lee et al., 2012*). To explain this striking genotype-phenotype correlation that is common to many trinucleotide repeat expansion disorders, a universal mechanism in which length-dependent somatic repeat expansion occurs toward a pathological threshold has been proposed (*Kaplan et al., 2007*). This mechanism provides a good explanation of the relationship between CAG repeat length and age-at-onset in HD very well as (1) the *HTT* CAG repeat shows increased repeat length mosaicism in the target brain region (*Telenius et al., 1994*; *Shelbourne et al., 2007*; *Mouro Pinto et al., 2020*), (2) somatic instability is repeat length-dependent (*Mouro Pinto et al., 2020*; *Ciosi et al., 2019*), and (3) the levels of repeat instability shows correlations with cell type-specific vulnerability and age-at-onset (*Shelbourne et al., 2007*; *Mouro Pinto et al., 2020*; *Swami et al., 2009*). In addition, somatic repeat instability of an expanded *HTT* CAG repeat appears to play a major role in modifying HD since our genome-wide association (GWA) studies have revealed that the majority of onset modification signals represent instability-related DNA repair genes (*Genetic Modifiers of Huntington's DiseaseConsortium, 2015*; *Genetic Modifiers of Huntington's Disease (GeM-HD) Consortium, 2019*; *Hong et al., 2021*; *Lee et al., 2017*). Together, these data support the critical importance of CAG repeat length and somatic instability in determining the timing of HD onset.

Recent large-scale genetic analyses of HD subjects have revealed that different DNA repeat sequence polymorphisms have an impact on age-at-onset. Most HD subjects carry an uninterrupted glutamine-encoding CAG repeat followed by a glutamine-encoding CAA-CAG codon doublet (referred to as a canonical repeat) (*Ciosi et al., 2019*; *Genetic Modifiers of Huntington's Disease (GeM-HD) Consortium, 2019*; *Wright et al., 2019*). However, expanded CAG repeats lacking the CAA interruption (loss of interruption) or carrying two consecutive CAA-CAG (duplicated interruption) (*Ciosi et al., 2019*; *Genetic Modifiers of Huntington's Disease (GeM-HD) Consortium, 2019*; *Wright et al., 2019*) also exist (*Figure 1—figure supplement 1*). Surprisingly, the age-at-onset of HD subjects carrying loss of interruption or duplicated interruption is best explained by the length of their uninterrupted CAG repeat, not the encoded polyglutamine length (*Ciosi et al., 2019*; *Genetic Modifiers of Huntington's Disease (GeM-HD) Consortium, 2019*; *Wright et al., 2019*). Still, age-at-onset of loss of interruption and duplicated interruption was not fully accounted for by uninterrupted CAG repeat, suggesting additional effects of non-canonical repeats. Nevertheless, these human genetics data indicate that introducing CAA interruption(s) into the *HTT* CAG repeat to reduce the length of the uninterrupted repeat is a potential therapeutic avenue to delay the onset of HD. Importantly, a genome engineering technology called base editing (BE) was recently developed, permitting the C-to-T conversion (cytosine base editor) or A-to-G conversion (adenine base editor) (*Komor et al., 2016*; *Nishida et al., 2016*; *Gaudelli et al., 2017*; *Levy et al., 2020*), where cytosine base editor could in principle be applied to convert CAG codons to CAA to shorten the uninterrupted CAG repeat without altering polyglutamine length or introducing different amino acids. In view of the strong human genetic evidence for the role of the uninterrupted CAG repeat length in determining HD onset (*Genetic Modifiers of Huntington's Disease (GeM-HD) Consortium, 2019*), we have conceived BE strategies of converting CAG codons to CAA within the repeat and evaluated their feasibility in HD.

## Results
### Effects of CAG-CAG interruption on age-at-onset in HD patients

Previously, we and others reported that most HD subjects carry canonical repeats comprising an uninterrupted expanded CAG repeat followed by CAA-CAG (*Ciosi et al., 2019*; *Genetic Modifiers of Huntington's Disease (GeM-HD) Consortium, 2019*; *Wright et al., 2019*). Although infrequent, uninterrupted CAG repeats followed by (1) no CAA-CAG (loss of interruption; 0.23% in our previous GWA data) and (2) two CAA-CAG codon doublets (duplicated interruption; 0.76% in our previous GWA data) also exist (*Genetic Modifiers of Huntington's Disease (GeM-HD) Consortium, 2019*). In

HD subjects carrying loss of interruption, the length of the CAG repeat and polyglutamine segment are identical. However, the polyglutamine length is greater by 2 and 4, respectively, compared to the CAG repeats in canonical repeat and duplicated interruption (*Figure 1—figure supplement 1*). Since canonical repeat, loss of interruption, and duplicated interruption with the same uninterrupted CAG repeat lengths have different polyglutamine sizes, they have provided a powerful tool to investigate the relative importance of the CAG repeat in DNA vs. polyglutamine in protein in determining onset age. For example, if polyglutamine length played an important role in determining age-at-onset, onset of loss of interruption and duplicated interruption carriers, who respectively have two fewer and two more glutamines compared to canonical repeat carriers, would be significantly later and earlier compared to canonical repeat carriers with the same uninterrupted CAG repeats (*Figure 1—figure supplement 2*). In stark contrast to these predictions, the onset ages of loss of interruption or duplicated interruption carriers are best explained by their respective CAG repeat sizes, not polyglutamine length (*Figure 1—figure supplement 2*). Furthermore, age-at-onset of duplicated interruption carriers is significantly delayed compared to that of loss of interruption carriers with the same uninterrupted CAG repeat size even though duplicated interruption encode four more glutamines than loss of interruption (Student's t-test p-value, 1.007E-12) (*Figure 1—figure supplement 2*). Together, these data indicate that age-at-onset in HD is determined primarily by the length of uninterrupted CAG repeat, but there may also be additional effects of different CAA interruption structures since the CAG repeat length does not fully explain age-at-onset in loss of interruption and duplicated interruption carriers (*Figure 1—figure supplement 2*; *McAllister et al., 2022*). Therefore, we performed least square approximation to calculate the magnitudes of the additional effects of loss of interruption and duplicated interruption on age-at-onset. Briefly, we varied the individual CAG repeat length to identify the repeat size that best explained the observed age-at-onset of carriers of these loss of interruption and duplicated interruption relative to canonical repeats. The age-at-onset of the loss of interruption carriers (n=21) was best explained when three CAGs were added to the true CAG repeat length (*Figure 1—figure supplement 3*) while the duplicated interruption carriers (n=69) behaved with respect to age-at-onset as if they had one less CAG than their true CAG repeat length (*Figure 1—figure supplement 3*). These data suggested that switching a canonical repeat to a duplicated interruption would delay onset by (1) shortening the uninterrupted CAG repeat by two CAG repeats and (2) conferring an additional effect equivalent to removing one CAG. For example, if a duplicated interruption is generated from a canonical repeat with 43 uninterrupted CAGs by converting the 42nd CAG to CAA using BE strategies, the age-at-onset is predicted to be delayed by approximately 12 years (*Figure 1D*), illustrating the robustness of therapeutic BE strategies.

## Cytosine base editors and gRNAs to convert CAG to CAA in the *HTT* CAG repeat

Recent advancements in genome editing technologies have led to the development of cytosine base editors that are capable of efficient C-to-T conversion (*Figure 2A*; *Komor et al., 2016*; *Nishida et al., 2016*; *Komor et al., 2017*; *Koblan et al., 2018*; *Thuronyi et al., 2019*). In principle, canonical repeat can be converted to duplicated interruption if cytosine base editors target the non-coding strand of the *HTT* CAG repeat (*Figure 2B*). In this study, we tested four cytosine base editors comprised of various cytosine deaminases and SpCas9 enzymes with different protospacer-adjacent motif (PAM) specificities to explore the feasibility of CAG-to-CAA conversion as a putative treatment strategy for HD. BE4 is the fourth-generation base editor which was engineered from BE3 to increase the editing efficiencies and decrease the frequency of undesired by-products (*Figure 2C*; *Komor et al., 2017*). BE4 exhibited high levels of C-to-T editing activity on the target sites harboring NGG PAMs (*Komor et al., 2017*). The activity window of BE4 is positions 4–8, counting from the PAM distal end of the spacer (where the PAM is positions 21–23) (*Figure 2B*; *Koblan et al., 2018*). We tested the BE4max (Addgene #112093) in this study, which is a codon-optimized version of BE4 with improved nuclear localization (*Koblan et al., 2018*). Due to the sparsity and lack of NGG PAM sites near and within the CAG repeat, CAG-to-CAA conversion using BE4 was expected to be somewhat limited. Therefore, we also explored engineered cytosine base editors containing SpCas9 variants that target an expanded range of PAM sequences, including SpCas9-NG (*Nishimasu et al., 2018*) and SpG (*Walton et al., 2020*; *Figure 2C*). Since these variants are capable of targeting sites with NGN PAMs, they might permit higher density targeting near or within the CAG repeat. The nucleotide preceding the

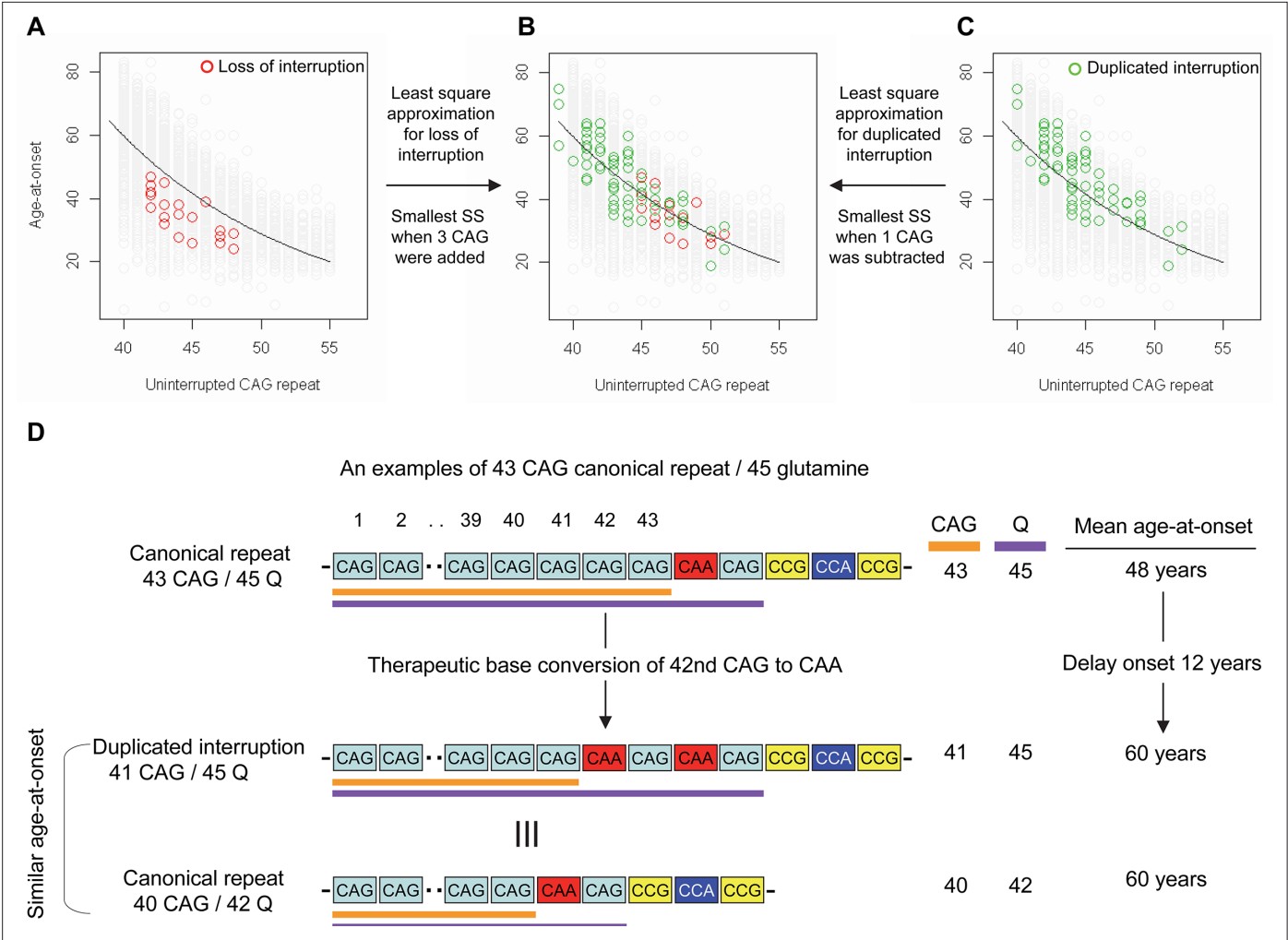

**Figure 1.** Effects of CAA interruption on Huntington's disease (HD) age-at-onset. (**A–C**) Least square approximation was performed to estimate the additional effects of loss of interruption (red circles in panels A and B; n=21) and duplicated interruption on age-at-onset (green circles in panels B and C; n=69). We varied the CAG length of HD participants carrying loss of interruption or duplicated interruption, and subsequently calculated sum of square to identify the CAG repeat that explained the maximum variance in age-at-onset of these allele carriers. Y-axis and X-axis represent age-at-onset and CAG repeat length, respectively. Gray circles and black trend lines respectively represent HD participants with canonical repeats and their onset-CAG relationship. SS means sum of square. (**D**) To illustrate the magnitude of the impact of a therapeutic base editing (BE) strategy of converting a canonical repeat to duplicated interruption by changing CAG to CAA, an example of a canonical repeat of 43 CAG (45 glutamine) with a mean observed onset of 48 years is displayed (n=564). In this example, therapeutic conversion of the 42nd CAG to CAA by BE would produce a duplicated interruption of 41 CAG (45 glutamine). Considering the additional effect of duplicated interruption in HD patients, a 41 CAG / 45 glutamine duplicated interruption would produce an onset similar to a canonical repeat allele of 40 CAG/42 glutamine, with a mean onset age of 60. Therefore, CAG-to-CAA conversion in HD subjects with 43 canonical repeat repeats could delay onset by 12 years.

The online version of this article includes the following figure supplement(s) for figure 1:

**Figure supplement 1.** The canonical repeat, loss of interruption, and duplicated interruption.

**Figure supplement 2.** Prediction of age-at-onset of loss of interruption and duplicated interruption carriers based on an assumption that polyglutamine determines onset.

**Figure supplement 3.** Least squares approximation to estimate the magnitude of additional effects of loss of interruption and duplicated interruption on age-at-onset.

target cytosine also affects the C-to-T conversion efficiency in cytosine base editors, especially when a G precedes the C (**Komor et al., 2016**; **Kim et al., 2017a**; **Gehrke et al., 2018**). Thus, engineered deaminase domains have been explored to improve C-to-T conversion in the GC contexts (**Nishida et al., 2016**; **Thuronyi et al., 2019**). For instance, an evolved CDA1-based BE4max variant (evoCDA1) showed substantially higher editing on GC targets (**Thuronyi et al., 2019**), which is relevant to the

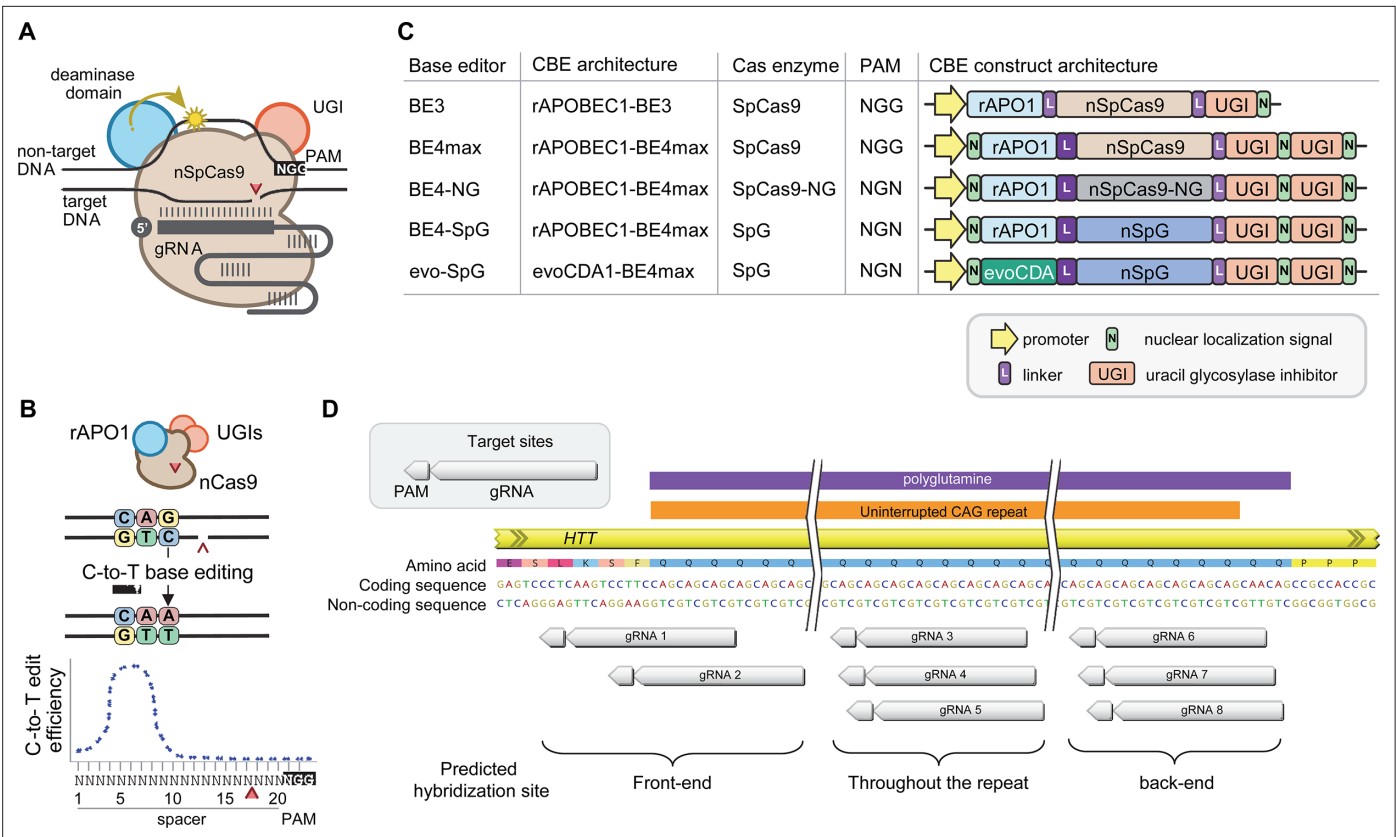

**Figure 2.** Cytosine base editors and guide RNAs (gRNAs) for CAG-to-CAA conversion in Huntington's disease (HD). (**A**) Constituents of base editing are displayed. (**B**) Schematic of cytosine base editors that can generate C-to-T conversion within a finite edit window at a fixed distance from the protospacer-adjacent motif (PAM). (**C**) Cytosine base editor variants described in the literature and used in this study (lower 4) are shown, including the evoCDA1-based SpG cytosine base editor that functions more efficiently in GC nucleotide contexts. PAM, gRNA, uracil glycosylase inhibitor (UGI), rat APOBEC1 deaminase domain (rAPO1), evolved CDA1 cytosine deaminase domain (evoCDA). (**D**) The target region, gRNAs, and expected hybridization sites of the eight gRNAs are shown.

The online version of this article includes the following figure supplement(s) for figure 2:

**Figure supplement 1.** The guide RNAs (gRNAs) for CBEs to convert CAG to CAA in Huntington's disease (HD).

nucleotide context on the non-coding strand of the *HTT* CAG repeat (CT<u>GC</u>TG). Therefore, we explored the use of BE4max-SpCas9, BE4max-SpCas-NG, BE4max-SpG, and evoCDA1-BE4max-SpG (henceforth referred to as BE4max, BE4-NG, BE4-SpG, and evo-SpG, respectively) (*Figure 2C*).

To achieve CAG-to-CAA conversion in the *HTT* CAG repeat, we designed three groups of gRNAs based on the sites of predicted hybridization (*Supplementary file 1*). Aiming at converting CAGs at the front-end of the repeat, gRNAs 1 and 2 were designed to hybridize with a region involving the upstream of the repeat and conventional NGG PAMs. The gRNAs 1 and 2 contain 10 and 2 non-CAG bases at the PAM-proximal ends, respectively (*Figure 2—figure supplement 1*). Considering the activity window of the BE4 (i.e. 13–17th nucleotide from the PAM), BE4max-gRNAs 1 and BE4max-gRNA 2 were predicted to convert the 1st/2nd and 4th/5th CAG to CAA, respectively (*Figure 2—figure supplement 1*). The gRNAs 3, 4, and 5 consist of CAGs (*Supplementary file 1*) and therefore, were predicted to hybridize throughout the *HTT* CAG repeat and potentially other CAG repeat-containing genes (*Figure 2—figure supplement 1*). The gRNAs 3, 4, and 5 were predicted to utilize NAA/NTG, NGA/NCT, and NGG/NGC PAMs, respectively (*Figure 2—figure supplement 1*). Lastly, gRNAs 6, 7, and 8 were designed to convert CAGs at the back-end of the repeat (*Supplementary file 1*). Available PAM sites for these gRNAs are NCT, NGC, and NTG (*Figure 2—figure supplement 1*). Considering the predicted gRNA-target hybridization sites and conversion windows, these three gRNAs might generate the duplicated interruption that is found in HD patients.

## Predominant CAG-to-CAA conversion without significant indels by BE strategies for HD

We then characterized 32 BE strategies (i.e. combinations of four cytosine base editors and eight gRNAs). We first determined whether BE strategies for HD produced indels. Since low BE efficiencies might result in proportionally low levels of indels leading to an underestimation of their frequencies, we used HEK293 cells, which showed high levels of BE efficiencies (*Fu et al., 2021*; *Xu et al., 2021*). Our MiSeq sequence analyses revealed that HEK293 cells carry two canonical repeats (16 and 17 CAGs) and showed approximately 10% of basal levels of indel (*Figure 3—figure supplement 1*), which reflects errors due to the difficulty in sequencing the CAG repeat. Nevertheless, transfection of plasmids for BE strategies did not significantly increase the levels of indels compared to cells without any treatment (Cell) or cells treated with empty vector (EV) (*Figure 3—figure supplement 1*). The lack of significant indel formation was quite expected because the cytosine base editors that we tested use nickases (*Figure 2C*; *Komor et al., 2017*; *Koblan et al., 2021*).

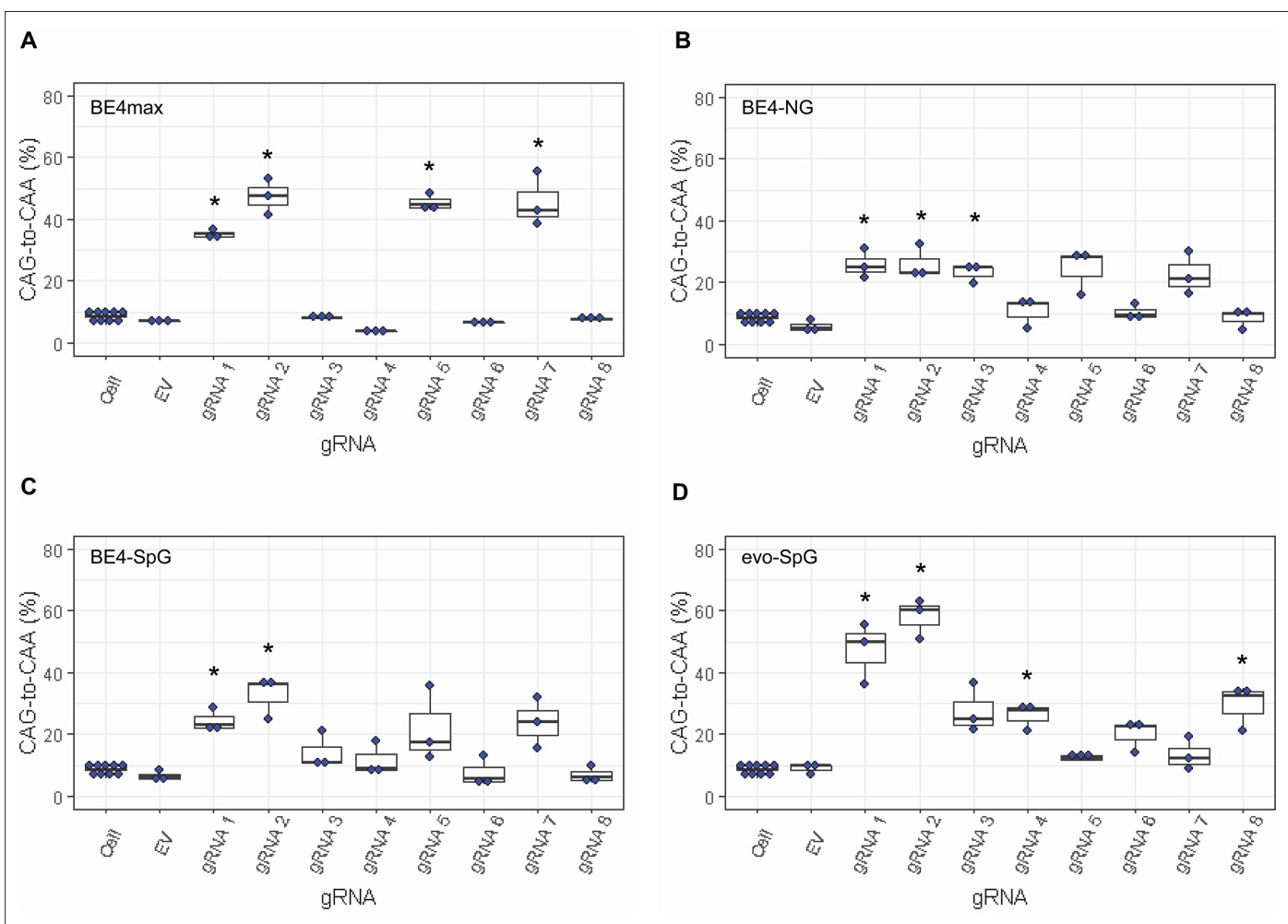

**Figure 3.** Levels of CAG-to-CAA conversion by base editing (BE) strategies. Only CAG-to-CAA conversion showed significantly increased levels over the baseline sequencing errors. Thus, we calculated the percentage of CAA in the cells that were treated with a combination of cytosine base editors (A, BE4max; B, BE4-NG; C, BE4-SpG; and D, evo-SpG) and guide RNAs (gRNAs) (n=3). HEK293 cells without any treatment (i.e. Cell) were combined (n=8) and plotted for each base editor. EV represents HEK293 cells treated with a base editor and empty vector for gRNA. *, significant by Bonferroni-corrected p-value<0.05 (eight tests for each base editor).

The online version of this article includes the following figure supplement(s) for figure 3:

**Figure supplement 1.** The lack of significant indels by base editing (BE) strategies.

**Figure supplement 2.** Types of base conversion by CBEs.

Since most sequence reads containing indels might be sequencing errors, we focused on sequence reads without indels to determine the types of base conversions. HEK293 cells without any treatment (Cell) or cells treated with EV showed low but detectable levels of CAG-to-CAA and CAG-to-TAG conversions (*Figure 3—figure supplement 2*), also reflecting sequencing errors. However, the levels of CAG-to-CAA conversion were significantly increased over baseline sequencing errors in cells treated with some BE strategies (*Figure 3*). For example, BE4max in combination with gRNAs 1, 2, 5, and 7 resulted in efficient CAG-to-CAA conversion (*Supplementary file 2*). Given the availability of the NGG PAMs, robust CAG-to-CAA conversion by gRNAs 1, 2, and 5 was somewhat anticipated for BE4max. However, high levels of CAG-to-CAA conversion by the BE4max-gRNA 7 combination (*Figure 3—figure supplement 2*) were unexpected because the anticipated hybridization site is not flanked by the NGG PAM that is required for the optimal activity of BE4max. The BE4-NG robustly produced CAG-to-CAA conversions with gRNAs 1, 2, and 3; although not significant, gRNAs 5 and 7 also generated high levels of CAG-to-CAA conversions (*Figure 3—figure supplement 2*). BE4-SpG with the combinations with gRNAs 1 and 2 resulted in significant levels of CAG-to-CAA conversions (*Figure 3—figure supplement 2*). Overall, the CAG-to-CAA conversion was higher in evo-SpG compared to other base editors; gRNAs 1, 2, 4, and 8 produced significant CAG-to-CAA conversions (*Figure 3—figure supplement 2*). These data indicated that our BE strategies primarily generated

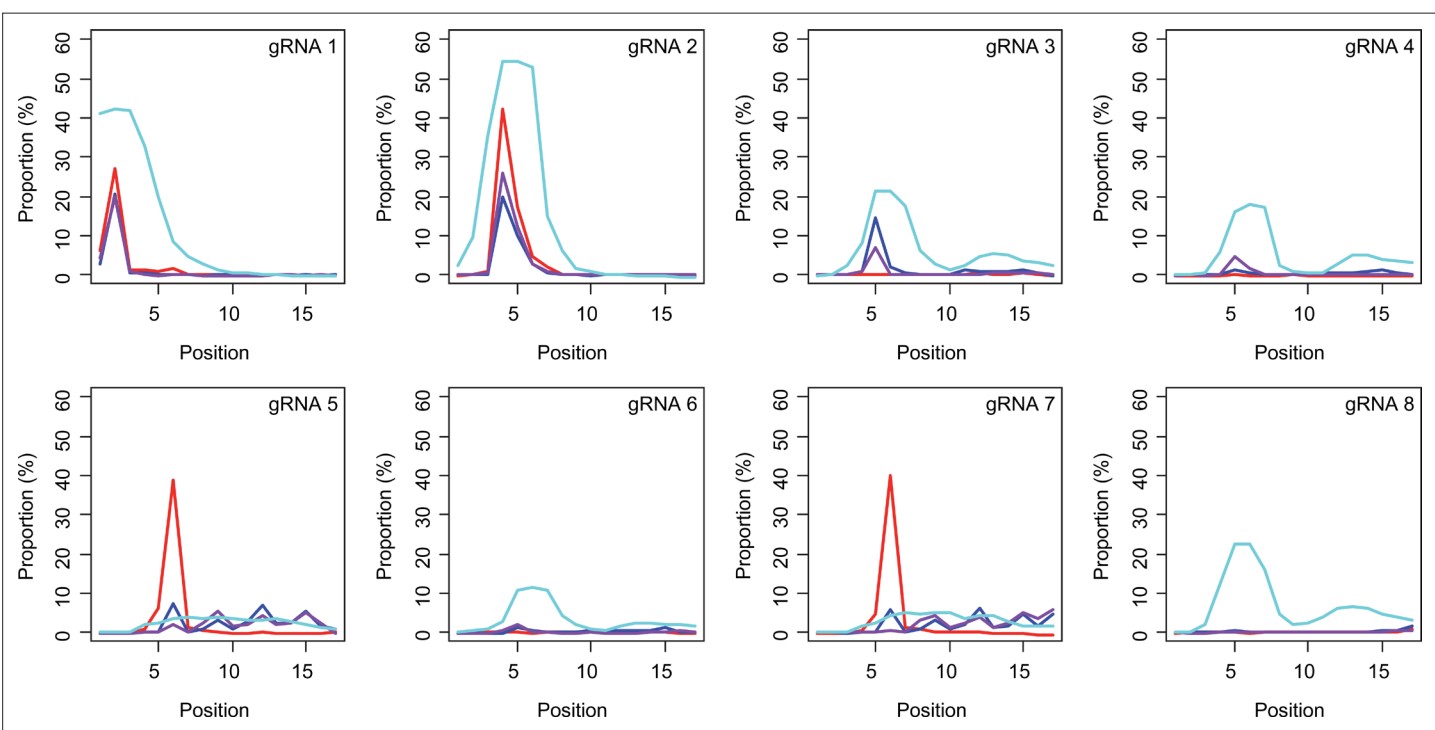

**Figure 4.** Sites of CAG-to-CAA conversion by base editing (BE) strategies. We calculated the mean percentage of sequence reads containing CAA at specific sites relative to all sequence reads (n=3). For example, 27.7% conversion at the 2nd CAG by BE4max-gRNA 1 (top left panel, red) means 27.7% of all sequence reads in HEK293 cells have CAA at the 2nd CAG. X-axis and Y-axis represent the position of the CAG and percent conversion. Each panel represents a tested gRNA. Plots were based on the mean of three independent transfection experiments in HEK293 cells after subtracting corresponding empty vector (EV)-treated cell data. Red, blue, purple, and cyan traces represent BE4max, BE4-NG, BE4-SpG, and evo-SpG, respectively.

The online version of this article includes the following figure supplement(s) for figure 4:

**Figure supplement 1.** Potential explanations for unexpected conversion sites.

**Figure supplement 2.** The levels of duplicated interruption alleles produced by base editing (BE) strategies.

**Figure supplement 3.** The levels of sequence read containing both duplicated interruption and CAG-to-CAA conversions at other sites.

**Figure supplement 4.** Transfection efficiency.

**Figure supplement 5.** The levels of multiple conversions.

**Figure supplement 6.** Off-target conversions on other polyglutamine disease genes.

**Figure supplement 7.** Characterization of differentiated neurons from a patient-derived induced pluripotent stem cell (iPSC).

CAG-to-CAA conversion without significant indel formation. Patterns of conversions also indicated that sites with NGG PAMs (gRNAs 1, 2, and 5) permitted the highest levels of CAG-to-CAA conversion for BE4max and other cytosine base editors with relaxed PAM specificities.

## Sites of CAG-to-CAA conversion

Subsequently, we determined conversion sites for different BE strategies. The patterns of conversion sites were similar for BE4max, BE4-NG, and BE4-SpG in gRNA 1, showing the most conversion at the second CAG with decreased levels of conversion at the first CAG (*Figure 4A*, *Supplementary file 3*). In contrast, evo-SpG-gRNA 1 combination showed higher editing efficiencies with the maximum conversion at the second CAG with comparable levels of conversions at the 1st and 3rd CAGs (*Figure 4A*, *Supplementary file 3*). The gRNA 2 showed similar patterns as gRNA 1 except that conversion sites were shifted to the right; the highest conversion occurred at the 4th CAG by BE4max, BE4-NG, and BE4-SpG (*Figure 4B*, *Supplementary file 3*).

The gRNAs 3 and 4, which were designed to hybridize throughout the CAG repeat, did not generate CAG-to-CAA conversion in combination with BE4max (*Figure 4C and D*, *Supplementary file 3*) because of the lack of an NGG PAM. Although modest, BE4-NG and BE4-SpG converted the 5th CAG to CAA (*Figure 4C and D*, *Supplementary file 3*), potentially due to the possibility that NAA (gRNA 3) and NGA (gRNA 4) PAMs supported the BE activity for BE4-NG and BE4-SpG. The gRNAs 3 and 4 produced higher levels of CAG-to-CAA conversion in evo-SpG again (*Figure 4C and D*; cyan), and interestingly, CAG-to-CAA conversions were not limited to the 5th CAG (*Supplementary file 3*). The gRNA 5 with BE4max efficiently converted the 6th CAG (*Figure 4E*; red), which was unexpected; conversions by other base editors were lower but widespread throughout the repeat.

BE strategies designed to convert CAGs at the back-end of the repeat were tested using gRNA 6, 7, and 8. Although less robust, the patterns of conversion by gRNA 6 (*Figure 4F*) were similar to those of gRNA 4 (*Figure 4D*). Since only one nucleotide is different between gRNA 6 and gRNA 4, it appeared that gRNA 6 behaved like gRNA 4 with one mismatch, favoring the NGA PAM instead of the less optimal NCT PAM (*Figure 4—figure supplement 1*). The same explanation might account for the similar patterns of conversion sites for gRNA 7 (*Figure 4G*) and gRNA 5 (*Figure 4E*); efficient conversion at the 6th CAG by BE4max-gRNA 7 might be due to the interaction of gRNA 7 at the target site of gRNA 5 (with one mismatch) in favor of the NGG PAM (*Figure 4—figure supplement 1*). The gRNA 8 generated CAG-to-CAA conversions only in evo-SpG. Although this group of gRNAs was designed to hybridize with the back-end of the CAG repeat, higher levels of conversion were observed at the front-end CAGs and throughout the repeat (*Figure 4F–H*). These results suggest that one PAM distal mismatch might be tolerated by base editors in favor of targets sites harboring more robust PAMs. Also, our data revealed that as expected, BE4max is highly dependent on the NGG PAM, resulting in CAG-to-CAA conversion at specific CAG sites, while evo-SpG is more efficient in conversion leading to broader targeting due to its relaxed PAM requirement.

## Generation of duplicated interruption by BE strategies

Next, we determined the levels of duplicated interruption in the same HEK293 cell MiSeq data. BE4max and evo-SpG did not produce significant amounts of the duplicated interruption that is found in humans. However, BE4-NG and BE4-SpG produced modest but significant levels of duplicated interruption in combinations with gRNAs 5 and 7 (0.5–1% increase over the basal levels) (*Figure 4—figure supplement 2*). Modest levels of duplicated interruption compared to conversions at other sites might be due to the lack of NGG PAM at the specific site (approximately 18 nucleotides upstream of CAA-CAG interruption). We also observed that gRNAs 5 and 7 relatively increased the number of sequence reads containing both duplicated interruption and CAG-to-CAA conversions at other sites (*Figure 4—figure supplement 3*), indicating that CTG trinucleotides on the non-coding strand of the repeat contributed to modest but widespread CAG-to-CAA conversion throughout the repeat. Similarly, CAG-to-CAA conversion was not confined to specific sites in evo-SpG in combination with gRNAs 3–8 as duplicated interruption generated by these strategies also contained CAG-to-CAA conversions at other sites (*Figure 4—figure supplement 3*). Since increased conversion efficiency in evo-SpG could not be explained by the transfection efficiency (*Figure 4—figure supplement 4*), these data indicate that evo-SpG has a significantly wider conversion window. In agreement with this,

the most frequent number of conversions in a given sequence read by evo-SpG was greater than that of other base editors (*Figure 4—figure supplement 5*, *Supplementary file 4*).

## Evaluation of off-target effects

We then evaluated the levels of off-target conversions using Off-Spotter. The gRNAs 1 and 2 showed relatively smaller numbers of predicted off-targets due to unique sequences near the PAMs (*Supplementary file 5*). As expected, gRNAs that were designed to hybridize throughout the CAG repeat showed increased numbers of predicted off-targets. Similarly, gRNAs to convert CAG at the back-end of the repeat showed larger numbers of predicted off-targets, potentially due to the fact that unique sequences are distal to the PAMs. Subsequently, we performed two sets of follow-up off-target validations. For gRNAs 1 and 2, we experimentally evaluated predicted off-targets focusing on protein-encoding genes; one and four genes were predicted off-target sites for gRNA 1 and gRNA 2, respectively, and all showed low levels of conversion compared to on-target (*Supplementary file 6*). We also characterized the levels of off-target conversion in other CAG repeat-containing genes focusing on eight polyglutamine disease genes (*Supplementary file 7*). As predicted, gRNAs 1 and 2 showed low-level conversions in the CAG repeats of other polyglutamine disease genes in general (*Figure 4—figure supplement 6*, *Supplementary file 8*). In contrast, gRNAs 3–8 produced variable but higher levels of conversion in some polyglutamine disease genes depending on the availability of preferred PAMs (*Figure 4—figure supplement 6*).

## Allele specificities and molecular outcomes of candidate BE strategies

Subsequently, we evaluate the levels of allele specificity of candidate BE strategies (BE4max-gRNA 1 and BE4max-gRNA 2) in patient-derived induced pluripotent stem cells (iPSC carrying 41 CAG CR) (*Shin et al., 2022a*; *Shin et al., 2022b*) and differentiated neurons (*Figure 4—figure supplement 7*). Then, gRNA 1 and gRNA 2 produced modest CAG-to-CAA conversion on both mutant and normal *HTT* (approximately <3%; *Supplementary file 9*). Overall, low conversion efficiencies by transfection and transduction of AAV (adeno-associated virus; data not shown) represent difficulties in delivery in these cell types, (*Shin et al., 2022a*; *Duong et al., 2019*) posing a challenge to determining the levels of allele specificity of BE strategies. To overcome these technical difficulties, we developed an HEK293 clonal line carrying an expanded *HTT* CAG repeat by replacing one of normal CAG repeats with a 51 CAG canonical repeat (namely HEK293-51 CAG) (*Figure 5—figure supplement 1*). A candidate BE strategy (i.e. BE4max-gRNA 1) did not increase the levels of in-frame insertion/deletion in the mutant or normal *HTT* repeat (*Figure 5B and C*). Subsequent analysis revealed that a candidate BE strategy BE4max-gRNA 1 produced high levels of CAG-to-CAA conversions on both expanded and non-expanded repeats (*Figure 5D*). Although very modest, conversion was significantly higher in the non-expanded repeat (uncorrected p-value, 0.04996), which can be explained by slightly reduced conversion on the mutant *HTT* due to higher GC content in the expanded CAG repeat. However, the candidate BE strategies did not alter huntingtin protein levels (*Figure 5E and F*) at the time of treatment, supporting the safety of candidate BE strategies. We also performed RNAseq analysis to identify genes whose expression levels were altered by candidate BE strategies in HEK293 cells. Candidate strategies such as BE4max-gRNA 1 and BE4max-gRNA 2 produced significant on-target CAG-to-CAA conversions (*Figure 6A*), but the levels of *HTT* mRNA were not altered by either treatment (*Figure 6—figure supplement 1*). In addition, RNAseq data analysis showed that neither BE strategy induced significant gene expression changes in any genes (false discovery rate, 0.05) (*Figure 6—figure supplement 1*). When comparing all HEK293 samples treated with either BE strategies (n=8) to those treated with EV (n=4), the shape of volcano plot mimicked random sample comparison (*Figure 6B and C*), implying the lack of impacts of candidate BE strategies on transcriptome.

## Effects of base conversion on the CAG repeat instability in vivo

The limited cargo capacity of AAV has been circumvented by the intein-split base editor, and the feasibility of BE strategies targeting non-repetitive sequences has been demonstrated in mouse models of human diseases (*Levy et al., 2020*; *Villiger et al., 2018*; *Koblan et al., 2021*). Taking advantage of the split BE system, we determined whether a candidate HD BE strategy could target the CAG repeat and result in a decrease in somatic repeat expansion, which was hypothesized to be the major disease driver (*Kaplan et al., 2007*). Since striatal and liver repeat instability share certain underlying

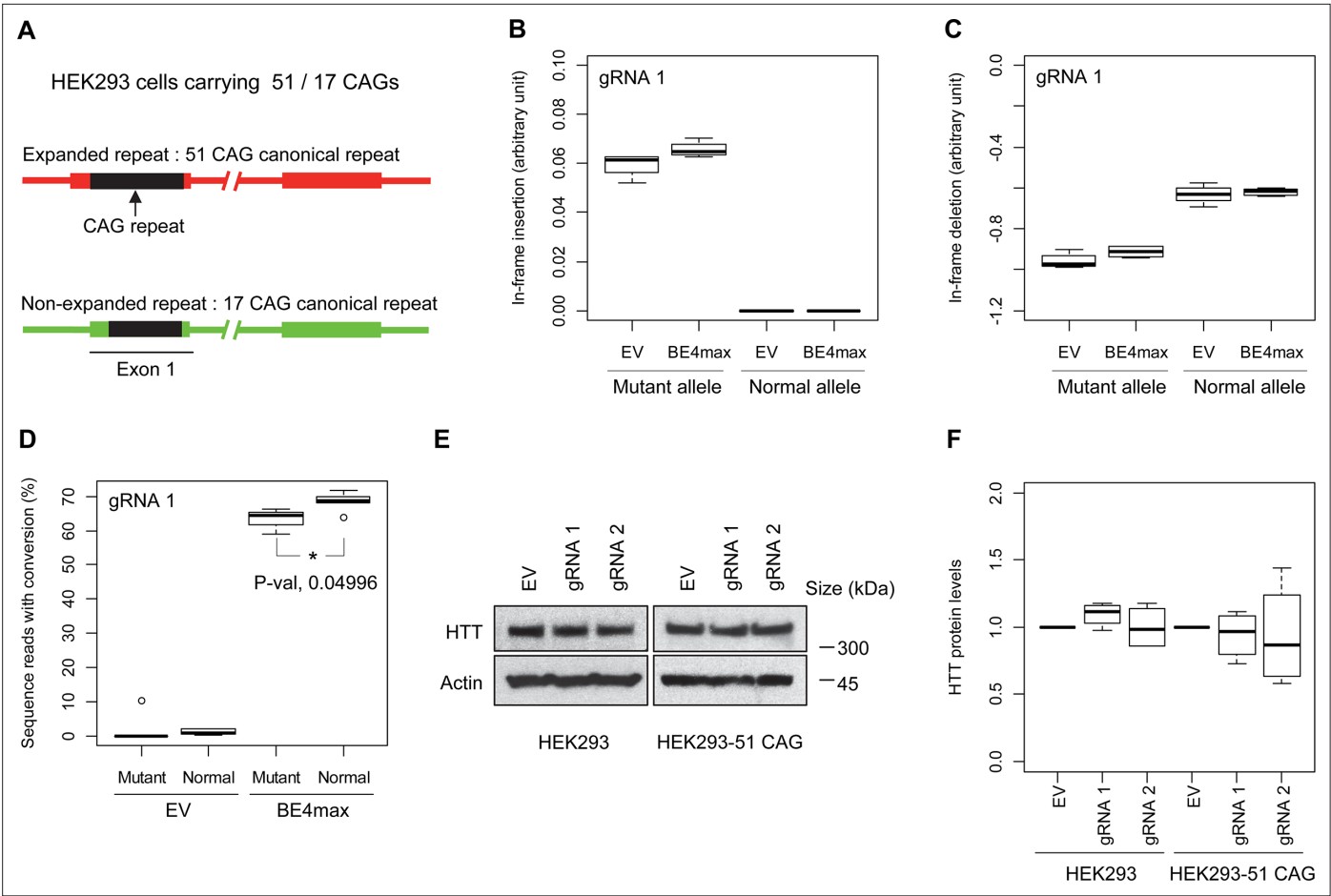

**Figure 5.** Allele specificity and molecular outcomes of candidate base editing (BE) strategies. (**A**) To overcome the limitations of patient-derived induced pluripotent stem cell (iPSC) and differentiated neurons, we developed HEK293 carrying an adult-onset CAG repeat by replacing one of the normal repeats with 51 canonical CAG (namely HEK293-51 CAG). Red and green bars represent respectively mutant and normal *HTT* in HEK293-51 CAG cells. (**B and C**) The HEK293-51 CAG cells were treated with BE4max-gRNA 1 and analyzed to determine the levels of in-frame insertion (**B**) and in-frame deletion (**C**) at the time of treatment (n=4). (**D**) The HEK293-51 CAG cells were treated with the gRNA 1 and analyzed by MiSeq to determine the levels of allele specificity. Conversion efficiency on the Y-axis indicates the percentage of sequence reads containing the CAG-to-CAA conversion at the target site (n=3). * represents uncorrected p-value<0.05 by Student's t-test. (**E**) Original HEK293 cells and HEK293-51 CAG cells were treated with empty vector (EV), or candidate BE strategies (BE4max-gRNA 1 and BE4max-gRNA 2) and subjected to immunoblot analysis; representative blot is shown in panel E (n=3). (**F**) Four independent experiments were performed, and we performed one-sample t-test to determine whether BE-treated cells show different total HTT protein levels compared to EV-treated cells (n=4). Nothing was significant by p-value<0.05.

The online version of this article includes the following source data and figure supplement(s) for figure 5:

**Source data 1.** Unedited original images of the western blot analysis in *Figure 5E*.

**Source data 2.** Uncropped images with labels of the western blot analysis in *Figure 5E*.

**Figure supplement 1.** Validation of HEK293-51 CAG cells.

**Figure supplement 1—source data 1.** Unedited original images of *Figure 5—figure supplement 1*.

**Figure supplement 1—source data 2.** Uncropped images with labels of *Figure 5—figure supplement 1*.

mechanisms (*Mangiarini et al., 1997*; *Manley et al., 1999*; *Kovtun and McMurray, 2001*; *Kennedy et al., 2003*; *Kovalenko et al., 2012*; *Pinto et al., 2013*; *Ament et al., 2017*; *Loupe et al., 2020*), and in vivo delivery might be more efficient in the liver compared to the brain (*Levy et al., 2020*), we used AAV9 to evaluate a candidate BE strategy in the liver. As expected, somatic CAG repeat expansion index in the liver of HD knock-in mice carrying around 110 CAGs showed a positive correlation with the inherited CAG repeat length (as represented in tail DNA) and the age of mice (*Figure 7A and B*; *Pearson et al., 2005*; *Mouro Pinto et al., 2020*; *Wheeler et al., 1999*; *Kacher et al., 2021*; *Kennedy and Shelbourne, 2000*; *Lee et al., 2011*). Unfortunately, we could not determine the

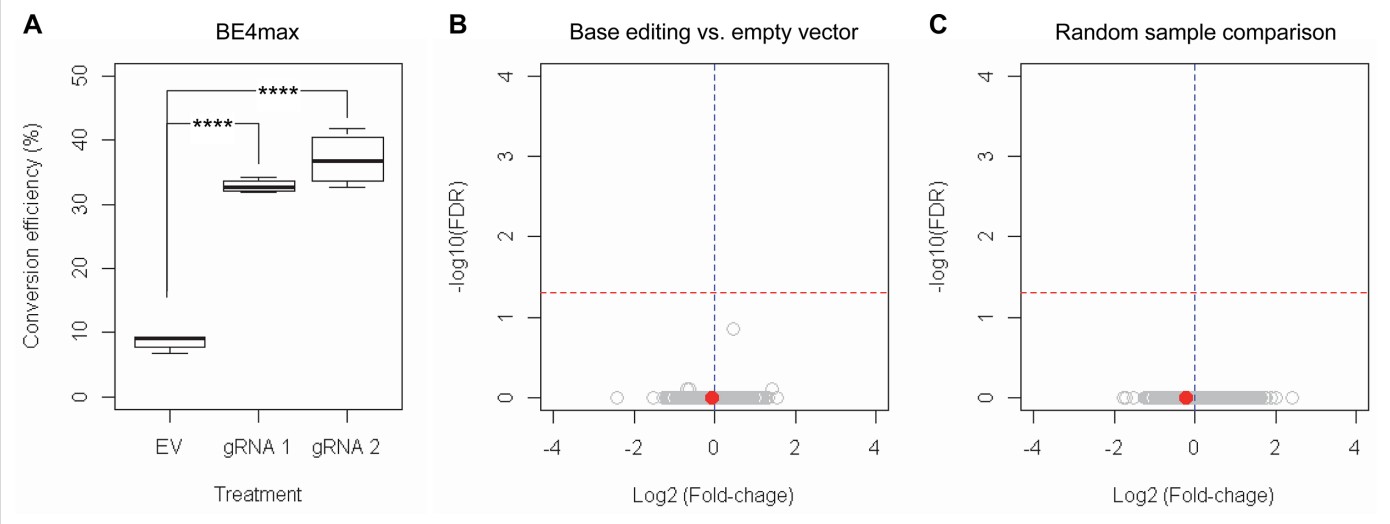

**Figure 6.** RNAseq analysis of base editing (BE) strategies confirms the lack of transcriptome alternation. (**A**) HEK293 cells were treated with empty vector (EV) or candidate BE strategies such as BE4max-gRNA 1 (gRNA 1) and BE4max-gRNA 2 (gRNA 2) for RNAseq analysis. MiSeq analysis was also performed to judge the levels of CAG-to-CAA conversion. ****, p-value<0.0001 by Student's t-test (n=4). (**B**) Confirming the lack of significantly altered genes in BE4max-gRNA 1 or BE4max-gRNA 2, we compared all BE-treated samples (n=8) with all EV-treated samples (n=4) to increase the power in the RNAseq differential gene expression analysis. Each circle in the volcano plot represents a gene analyzed in the RNAseq; *HTT* is indicated by a filled red circle. A red horizontal line represents false discovery rate of 0.05, showing that none was significantly altered by candidate BE strategies. (**C**) We also compared two groups of randomly assigned samples (six samples vs. six samples) to understand the shape of the volcano plot when there were no significant genes.

The online version of this article includes the following figure supplement(s) for figure 6:

**Figure supplement 1.** The lack of significant alterations in gene expression by BE4max-gRNA 1 and BE4max-gRNA 2.

modified sequence in the treated mice by sequencing because of (1) very long CAG repeats in these mice, (2) modest levels of base conversion, and (3) high levels of errors when sequencing the CAG repeat. However, when the effects of the tail CAG repeat size and age of mice were corrected, retro-orbital injection of AAV9 for split cytosine base editor and gRNA 2 significantly decreased the levels of repeat expansion (*Figure 7C*; p-value, 1.78E-6). Nevertheless, the expansion index in treated and control mice was largely overlapping (*Figure 7A*), suggesting that the effects of BE treatment were very modest. We speculate that (1) insufficient dosage due to difficulty in producing high titer viral package for big cargo (i.e. 5 KB) (*Levy et al., 2020*; *Wu et al., 2010*), (2) limited delivery (*Carvalho et al., 2017*), and/or (3) difficulty in targeting the very long CAG repeat resulted in modest effects. Given those limitations, we also analyzed a mouse model containing interrupted repeat to determine the maximum effects of the interruption on the repeat expansion. HD knock-in mice carrying 105 interrupted CAG repeat (https://www.jax.org/strain/027418) showed complete loss of repeat expansion compared to 105 CAG uninterrupted repeat mice (https://www.jax.org/strain/027417; *Figure 7D and E*), suggesting that CAA interruption could completely suppress the most important disease modifier (i.e. CAG repeat expansion).

## Discussion

Recent advances in genome engineering provide powerful tools to interrogate the relationships among genes, functions, and diseases. For example, CRISPR-Cas9-based editing approaches have revolutionized the investigation of genes of interest and also have begun to be applied to humans to treat diseases (*Doudna and Charpentier, 2014*; *Hsu et al., 2014*; *Stadtmauer et al., 2020*; *Gillmore et al., 2021*; *Wang et al., 2021*). BE, which can convert a single nucleotide to another, represents a newly developed and highly versatile genome engineering technology (*Komor et al., 2016*; *Gaudelli et al., 2017*). BE has advantages over other genome engineering approaches with respect to safety and clinical applicability. BE employing nickase Cas9 does not intentionally create double-stranded DNA breaks (DSBs) (*Komor et al., 2016*; *Gaudelli et al., 2017*), minimizing potential adverse effects.

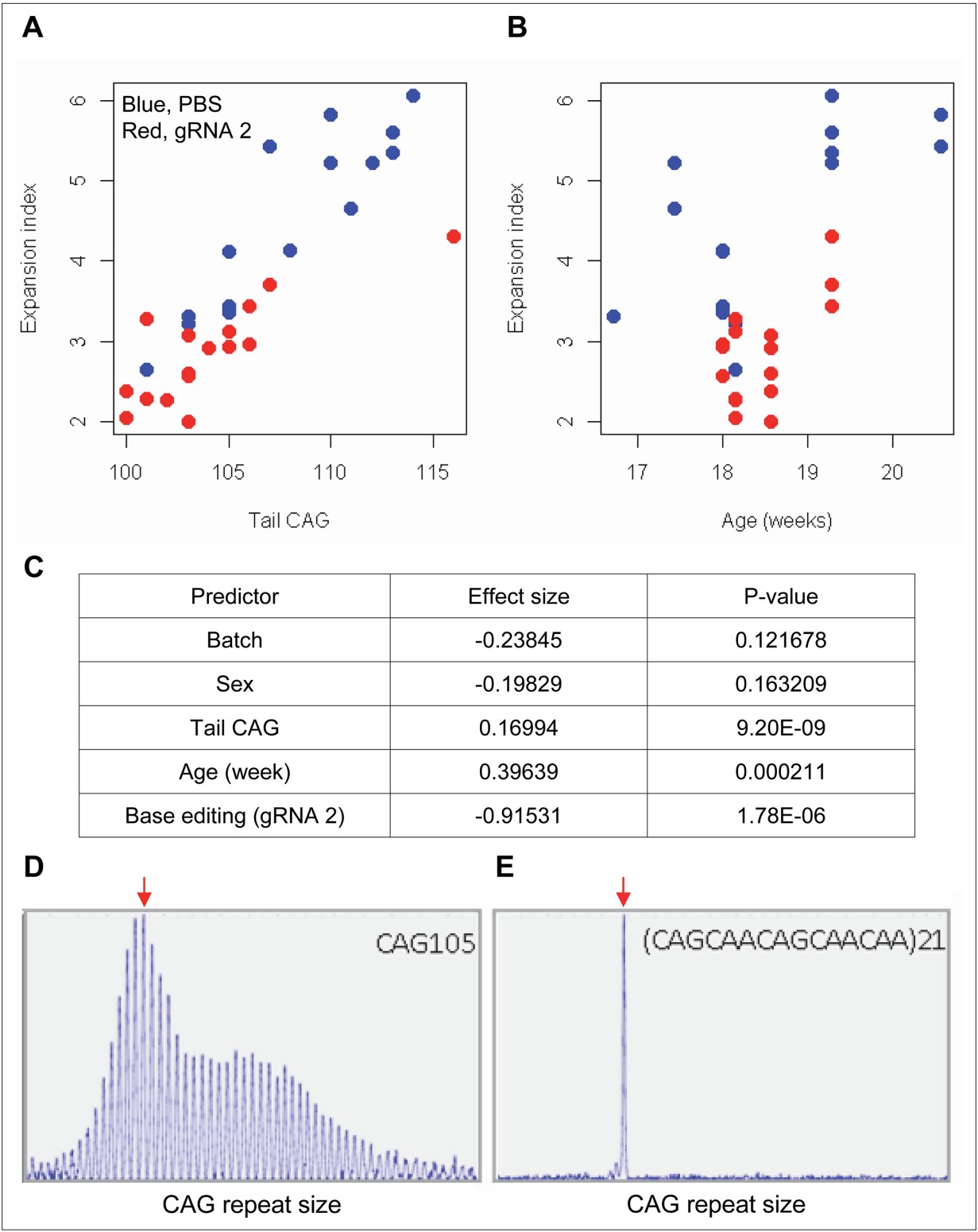

**Figure 7.** Impacts of CAA interruption on CAG repeat instability. (A–C) DNA samples (liver and tail) of base editing (BE)-treated (guide RNA [gRNA] 2, red; n=16) or PBS-treated (PBS, blue; n=15) were analyzed to quantify somatic repeat expansion. We performed linear regression analysis to model the levels of repeat expansion as a function of treatment, CAG repeat in tail (**A**), age (**B**), and with other covariates (i.e. experimental batch, sex, tail CAG, and age). Summary of the statistical analysis is summarized in the panel C. (**D and E**) To determine the maximal impacts of CAA interruption on the repeat expansion, Huntington's disease (HD) knock-in mice carrying CAA-interrupted repeats were analyzed. Liver samples of 105 uninterrupted CAG repeat (**D**) and interrupted repeat (**E**) were analyzed at 5 months. Representative fragment analysis is displayed. Red arrows indicate the modal alleles representing inherited CAG repeats; peaks at the right side of the modal peaks (red arrows) represent expanded repeats.

Also, BE with low off-targeting is being actively developed to increase safety (*Neugebauer et al., 2023*). The majority of well-characterized disease-causing mutations are point mutations, and therefore many genetic disorders can be addressed by BE strategies (*Rees and Liu, 2018*). The robustness of BE has been demonstrated in models of genetic disorders caused by point mutations (*Villiger et al., 2018*; *Koblan et al., 2021*; *Rees and Liu, 2018*; *Newby et al., 2021*), and the first human trial employing BE has already been started (*Kingwell, 2022*; *Eisenstein, 2022*). However, many human disorders are caused by other types of mutations, such as expansions of DNA repeats (*Orr and Zoghbi, 2007*; *Pearson et al., 2005*; *McMurray, 2010*) for which BE may not seem like an ideal tool. In contrast to this commonly held notion, we show that BE strategies could also address diseases that are caused by expanded repeats, broadening their target space and applicability.

In HD, multiple studies have shown that the uninterrupted CAG repeat length in the *HTT* gene, not the polyglutamine length in huntingtin protein, determines age-at-onset (*Ciosi et al., 2019*; *Genetic Modifiers of Huntington's DiseaseConsortium, 2015*; *Wright et al., 2019*). Age-at-onset of HD subjects carrying duplicated interruption not only supports this notion directly, but also points to novel therapeutic strategies. For example, converting CAG to CAA would decrease the length of uninterrupted CAG repeat without changing the length of polyglutamine or altering huntingtin protein. Indeed, our candidate BE strategies could shorten the length of uninterrupted CAG repeat by converting CAG to CAA at various sites in the CAG repeat without producing significant indels or off-target effects. In support, our candidate BE strategy modestly but significantly reduced the levels of CAG repeat expansion in mice, and HD knock-in mice carrying the CAA-interrupted repeats showed virtually zero repeat expansion. Given the role of the uninterrupted CAG repeat length as the most important disease determinant and a pivotal role for repeat instability in the modification of HD (*Genetic Modifiers of Huntington's DiseaseConsortium, 2015*; *Genetic Modifiers of Huntington's Disease (GeM-HD) Consortium, 2019*), our data support the therapeutic potential of CAG-to-CAA conversion BE strategies in HD.

Our data are relevant for a number of reasons. First, genetically supported targets significantly increase the success rate in clinical development (*Nelson et al., 2015*; *King et al., 2019*; *Hingorani et al., 2019*); our BE strategies were derived directly from human genetic observations in HD individuals, (*Ciosi et al., 2019*; *Genetic Modifiers of Huntington's Disease (GeM-HD) Consortium, 2019*; *Wright et al., 2019*) pointing to the uninterrupted CAG repeat length in *HTT* as the most direct therapeutic target in HD. Based on these human data, CAG-to-CAA conversion even near the 5'-end of the expanded CAG repeat may produce robust onset-delaying effects. In addition, if BE strategies are applied to fully penetrant 40 or 41 CAG canonical repeats, the repeats can become reduced penetrant (i.e. 36–39). Similarly, BE strategies may be able to convert some of the reduced penetrant CAG repeats (e.g. 36 and 37 CAG) to non-pathogenic (CAG<36), which can prevent the manifestation of the disease. Second, lessons from the recent huntingtin-lowering clinical trial (*Relations, 2021*; *Sheridan, 2021*) implied the importance of allele-specific approaches (*Shin et al., 2022a*; *Shin et al., 2022b*). Although CAG-to-CAA conversions in our experiments occurred on both mutant and normal *HTT*, our candidate BE strategies are expected to produce mutant allele-specific consequences. The amino acid sequence and levels of huntingtin were not altered, while the length of the uninterrupted CAG repeat was decreased. On the mutant allele, this shortening is expected to reduce the somatic instability, lowering its disease-producing potential. On the normal allele, inherited variation in the length of the CAG repeat has not been associated with an abnormal phenotype (*Lee et al., 2012*), so the shortening of the CAG repeat is expected to be benign. Importantly, the mutant allele-specific consequences can be achieved without relying on individual genetic variations beyond the CAG repeat. Various SNP-targeting allele-specific approaches have been proposed, (*Shin et al., 2022a*; *Shin et al., 2022b*; *Shin et al., 2016*; *Monteys et al., 2017*) but most of these can be applied only to a subset of the HD population depending on heterozygosity at the target site. Our BE strategies can achieve allele-specific consequences without targeting SNPs, and, therefore, can be applied to all HD subjects, representing a huge advantage over SNP-targeting allele-specific strategies. Lastly, BE with relaxed PAM requirements *Walton et al., 2020*; *Rees and Liu, 2018* has increased the applicability of BE, and our study has further expanded the target space of this powerful technology. BE is appropriately viewed as a tool to correct disease-causing point mutations or to modify gene expression by introducing early stop codons or altering splice sites (*Rees and Liu, 2018*; *Kuscu et al., 2017*; *Kim et al., 2017b*). However, our study demonstrates that base conversion can address disease-causing

repeat expansion mutations without involving DSB. The ramifications apply not only to HD but also to numerous other diseases that are caused by expansions of repeats (*Gusella and MacDonald, 2000*; *Ross, 2002*; *Di Prospero and Fischbeck, 2005*; *Orr and Zoghbi, 2007*; *Depienne and Mandel, 2021*), offering alternative therapeutic approaches for the repeat expansion disorders.

Although promising, hurdles must be overcome before CAG-to-CAA conversion BE strategies are applied to humans. The BE strategies that we evaluated did not robustly generate duplicated interruption that are found in humans, potentially due to the possibility that PAM at specific sites that are required to generate duplicated interruption did not sufficiently support the activity of cytosine base editors that we tested. Therefore, new cytosine base editors that can efficiently generate duplicated interruption will greatly facilitate the development of rational treatments for HD. Also, the inability to directly target putative toxic species such as RAN translation or exon 1A huntingtin fragment (*Yang et al., 2020*; *Neueder et al., 2017*; *Bañez-Coronel et al., 2015*; *Sathasivam et al., 2013*) may represent one of the limitations of our BE strategies for HD. Still, if the levels of those alternative toxic species are dependent on the length of uninterrupted CAG repeat, CAG-to-CAA conversion strategies may be able to ameliorate toxic species-mediated HD pathogenesis. Although BE strategies can address the primary disease driver in principle, they may not produce any significant clinical benefits if they are applied too late in the disease. As we previously speculated, the timing of treatment might have negatively impacted the outcomes of the first ASO *HTT*-lowering trial (*Shin et al., 2022a*; *Shin et al., 2022b*; *Relations, 2021*; *Sheridan, 2021*). Considering the evidence for significant levels of neurodegeneration at the onset of characteristic clinical manifestations (*Paulsen et al., 2008*), CAG-to-CAA conversion treatments may not produce any clinical improvements if applied late. With the expectation of mutant-specific consequences, we reason that BE strategies can be applied quite early without involving deficiency-related adverse effects (*Lopes et al., 2016*; *Rodan et al., 2016*; *Dietrich et al., 2017*; *Wang et al., 2016*) because CAG-to-CAA conversion is predicted to neither alter the amino acid sequence nor changes the expression levels of *HTT*. Regardless, these temporal aspects and safety features have to be determined. Finally, like other gene-targeting strategies, the development of effective delivery methods is critical for applying BE therapeutically. The expansion-decreasing effects of our initial AAV injection experiments, while significant, may have been limited compared to cell culture systems by inefficient delivery and difficulty in targeting the repeat sequence. For the successful application of BE strategies to human HD, efficient delivery methods will be critical.

Given the lack of effective treatments for HD and the premature terminations of highly anticipated *HTT*-lowering clinical trials such as GENERATION-HD1 (*Relations, 2021*) and VIBRANT-HD, aiming at the most relevant target is becoming increasingly important. Our data reveal relevant strategies for addressing the target strongly supported by human HD genetic data, the uninterrupted CAG repeat in *HTT*, therefore offer new opportunities for blocking the disorder at its cause. Currently, the ideal levels of CAG-to-CAA conversion that produce significant clinical benefits are unknown. A series of preclinical studies using relevant model systems may generate data that may shed light on the optimal conversion rate levels that are required to produce significant clinical benefits. In addition, developing base editors with high levels on-target gene specificity and minimal off-target effects is a critical aspect to address (*Li et al., 2022*). Also, adverse effects due to immunogenicity are issues that may hamper the application of BE strategies to humans. Thorough assessments of immune responses against BE strategies (e.g. development of antibody, B cell, and T cell-specific immune responses) and subsequent modification (e.g. immunosilencing) (*Ewaisha and Anderson, 2023*) will be critical to address immune response-associated safety issues of BE strategies. Although both great promise and significant hurdles exist for the clinical application of BE strategies, our data illustrating genetic rationale and demonstrating the proof-of-concept of this technology may contribute to developing a rational treatment for HD and, potentially, for other repeat expansion disorders.

# Materials and methods
## Study approval

This study analyzed only data from human subjects, not involving new recruitment. The original subject consents and the overall study were approved by the Mass General Brigham IRB and described previously (*Genetic Modifiers of Huntington's Disease (GeM-HD) Consortium, 2019*). Experiments involving mice were approved by the Mass General Brigham Institutional Animal Care

and Use Committee (protocol numbers, 2018N000220 and 2020N000135). Every effort was made to minimize suffering.

## Age-at-onset of HD subjects carrying loss of interruption or duplicated interruption

Detailed experimental procedures for sequencing of the *HTT* CAG repeat region and determination of the CAG repeat length are described previously (*Genetic Modifiers of Huntington's Disease (GeM-HD) Consortium, 2019*). We compared age-at-onset of HD subjects carrying loss of interruption or duplicated interruption to that of HD subjects carrying canonical repeats. Expected age-at-onset from CAG repeat length of canonical repeat was based on the onset-CAG regression model that we reported previously (*Lee et al., 2012*; *Genetic Modifiers of Huntington's DiseaseConsortium, 2015*; *Genetic Modifiers of Huntington's Disease (GeM-HD) Consortium, 2019*). For expected age-at-onset based on the polyglutamine length, the same regression model was modified by replacing CAG repeat length with CAG repeat length plus 2 because the glutamine length equals CAG+2 in canonical repeats.

## Least square approximation to estimate the additional effects of loss of interruption and duplicated interruption on age-at-onset

Carriers of loss of interruption and duplicated interruption showed slightly earlier and later onset age, respectively, compared to those with canonical repeat of the same uninterrupted CAG repeat lengths, suggesting that loss of interruption and duplicated interruption confer additional effects. Thus, we attempted to determine the levels of additional effects of loss of interruption and duplicated interruption that were not explained by the uninterrupted CAG size by taking a mathematical approach that is similar to least square approximation. Briefly, we predicted age-at-onset of HD subject carrying loss of interruption or duplicated interruption using our CAG-onset regression model for canonical repeat (*Lee et al., 2012*), and subsequently calculated the residual by subtracting predicted onset from observed onset age. We then calculated the sum of square for loss of interruption and duplicated interruption carriers based on the participant's true uninterrupted CAG repeat length using the following formula.

$$\text{Sum of squares} = \Sigma(\text{observed age} - \text{at} - \text{onset} - \text{predicted age} - \text{at} - \text{onset age})^2$$

Subsequently, we gradually increased and decreased the CAG repeat length for loss of interruption and duplicated interruption carriers, respectively, and calculated sum of square again to identify CAG repeat size that generated the smallest sum of square value. The differences between true CAG and CAG repeat length that produced the smallest sum of square were considered as additional effects of loss of interruption and duplicated interruption on age-at-onset.

## Cell line

HEK293 cells were obtained from the ATCC (https://www.atcc.org/products/crl-1573); the identity of the cell line and mycoplasma contamination have not been tested. Cells were maintained in DMEM containing L-glutamine supplemented with 10% (vol/vol) FBS and 1% penicillin-streptomycin (10,000 U/ml) at 37°C and 5% $CO_2$. TrypLE Express (Life Technologies) was used to detach cells for sub-culture.

## gRNA cloning and transfection

PX552 vector (Addgene #60958) was digested using SapI (Thermo) and purified by gel purification (QIAquick Gel Extraction Kit). A pair of oligos for each gRNA were phosphorylated (T4 Polynucleotide Kinase, Thermo) and annealed by incubating at 37°C for 30 min, 95°C for 5 min, and ramping to 4°C. Annealed oligos were diluted (1:50) and ligated into the digested PX552 vector (T7 ligase, Enzymatic) and incubated at room temperature for 15 min. Then, transformation was performed (One Shot Stbl3, Invitrogen). The inserted gRNA sequences were confirmed by Sanger sequencing. For transfection, cells were seeded in six-well plates at approximately 65% confluence and treated with 1.66 μg of cytosine base editor and 0.7 μg of gRNA plasmids on the following day using Lipofectamine 3000 (Invitrogen) according to the manufacturer's protocol. Three days after transfection, cells

were harvested for molecular analysis. Genomic DNA was extracted using DNeasy Blood & Tissue kit (QIAGEN). AccuPrime GC-Rich DNA Polymerase (Invitrogen) was used to amplify a region containing the CAG repeat (35 cycles). PCR product was purified by PCR QIAquick PCR Purification Kit (QIAGEN) and subjected to MiSeq (Center for Computational and Integrative Biology DNA Core, Massachusetts General Hospital) and/or Sanger sequencing analysis (Center for Genomic Medicine Genomics Core, Massachusetts General Hospital). Primers for MiSeq sequencing were ATGAAGGCCTTCGAGTCCC and GGCTGAGGAAGCTGAGGA; primers for Sanger sequencing analysis were CAAGATGGACGG CCGCTCAG and GCAGCGGGCCCAAACTCA.

## MiSeq data analysis to determine indels and conversion types

Sequence data from the MiSeq sequencing were subject to quality control by removing sequence reads (1) with mean base Phred quality score smaller than 20, (2) showing the difference between forward and reverse read pair, (3) containing fewer than 6 CAGs, or (4) not involving the full primer sequences. Quality control-passed data revealed that HEK293 cells carry 16/17 CAG canonical repeats and therefore are expected to produce 18/19 polyglutamine segments. For quality control-passed sequence reads, we determined the proportion of sequence reads containing indels, revealing most indels were sequencing errors. Subsequently, we focused on sequence reads without indels to determine the types of conversion. For each sequence read (not including CAA-CAG interruption), we counted sequence reads containing CAA, CAC, CAG, CCG, CGG, CTG, AAG, GAG, and TAG trinucleotide to determine the types and levels of conversion.

## MiSeq data analysis of HEK293 cells treated with BE strategies to determine the sites of conversion

Sequence analysis revealed that BE strategies using cytosine base editors produced mostly CAG-to-CAA conversion. CAG-to-TAG conversion was detected in all samples regardless of BE strategies, suggesting that this type of conversion is also due to amplification/sequencing errors. Therefore, we focused on sequence reads of 16/17 CAG repeats containing only CAG or CAA to determine the sites of CAG-to-CAA conversion. Briefly, we recorded the sites of CAG-to-CAA conversion for each sequence read and summed the number of conversions at a given site for a given sample. Therefore, 30% CAG-to-CAA conversion at the second CAG means 70% and 30% of all sequence reads contain CAG and CAA at the second CAG position, respectively.

## Quantification of duplicated interruption and multiple conversion

The proportion of duplicated interruptions was determined from HEK293 cells treated with different combinations of cytosine base editors and gRNAs. Briefly, we counted sequence reads containing duplicated interruption and divided them by the number of all quality control-passed sequence reads to calculate the proportion of duplicated interruption. Similarly, we calculated the proportion of sequence reads containing duplicated interruption and CAG-to-CAA conversions at other sites. We also determined the levels of multiple CAG-to-CAA conversion for each BE strategy. For each sequence read in a sample, we counted the number of CAG-to-CAA conversions regardless of their locations to evaluate a distribution of numbers of multiple conversion for each sample. Since we counted the conversions regardless of their positions, multiple conversions do not necessarily mean consecutive conversions.

## Determination of the transfection efficiency

To determine the effects of transfection efficiency on patterns of BE, we transfected HEK293 cells with gRNA 2 with combinations of different base editors and performed cell staining. Transfected HEK293 cells were fixed with paraformaldehyde (4%) and permeabilized with Triton X-100 (0.5%). Then, cells were stained with DAPI (0.5 µM) and incubated for 30 min before being washed with PBS. The eGFP (enhanced green fluorescent protein) and DAPI (4',6-diamidino-2-phenylindole) images from eight areas in each well were captured using a fluorescence inverted microscope (Nikon Eclipse TE2000-U). The ImageJ analysis program was used to measure the size of a single cell expressing eGFP; we randomly selected 20 cells for each image and averaged their sizes to be used as a reference. We counted the number of pixels covered by eGFP-positive signals, and subsequently divided by the average cell size to obtain the number of eGFP-positive cells in each image. This was repeated with

the DAPI staining images. The percent transfected was calculated by dividing the number of eGFP-positive cells by that of DAPI-positive cells multiplied by 100.

## BE in HD patient-derived iPSC and differentiated neurons

An iPSC line carrying adult-onset CAG repeats (42 CAG) was derived from a lymphoblastoid cell line in our internal collection by the Harvard Stem Cell Institute iPS Core Facility (http://ipscore.hsci.harvard.edu/; *Shin et al., 2022a*; *Shin et al., 2022b*). HD iPSCs were dissociated into single cells with Accutase (STEMCELL Technologies) and plated on Matrigel-coated 24-well plate in mTeSR plus media containing CloneR (STEMCELL Technologies) to increase cell viability. The following day, cells at 60–70% confluence were transfected with 1.8 µg of BE4max and 0.6 µg of gRNA plasmids using Lipofectamine STEM (Invitrogen) according to the manufacturer's protocol. Cells were incubated at 37°C and 5% $CO_2$ for 5 days for sequencing analysis.

The same iPSC line was differentiated into neurons using a previously described method (*Fjodorova and Li, 2018*). Briefly, the iPSC line was plated on growth factor reduced Matrigel (Corning) in mTeSR Plus media (STEMCELL Technologies). When cells reached ~80% confluence, differentiation was initiated by switching to DMEM-F12/Neurobasal media (2:1) supplemented with N2 and retinol-free B27 (N2B27 RA−; Gibco). For the first 10 days, cells were supplemented with SB431542 (10 µM; Tocris), LDN-193189 (100 nM; StemGene), and dorsomorphin (200 nM; Tocris). SB431542 was removed from the media on day 5. Cells were maintained in N2B27 RA− supplemented with activin A (25 ng/ml; R&D) on day 9. On day 22, cells were split using Accutase (STEMCELL Technologies) and seeded on a poly-D-lysine/laminin plate with N2B27 media supplemented with BDNF and GDNF (10 ng/ml each; Peprotech). Media were changed the next day to facilitate neuronal maturation and survival. Cells were fed with new media every 2 days. For neuronal marker staining, cells were fixed, permeabilized, and blocked using the Image-iT Fix-Perm kit (Invitrogen). Subsequently, cells were stained by Anti-TUBB3 (tubline beta 3; Biolegend Inc, Cat# 801202) in a blocking solution overnight at 4°C. Then, cells were washed with PBS three times for 5 min, followed by incubation with Alexa Flour 594 secondary antibodies (Invitrogen) for 1 hr. Finally, cells were washed with PBS three times for 5 min and mounted with Vectorshield mounting medium with DAPI (Vector Laboratories). Images were captured by the Leica fluorescence microscope. Differentiated neurons were transfected with 1.8 µg of BE4max and 0.6 µg of gRNA plasmids using Lipofectamine 3000 according to the manufacturer's protocol. Cells were incubated at 37°C and 5% $CO_2$ for 7 days for sequencing analysis.

## Off-target prediction and experimental validation

Potential off-targets were predicted by the Off-Spotter (https://cm.jefferson.edu/Off-Spotter/) for eight BE strategies using the gRNA sequences. We allowed a maximum of four mismatches to identify potential off-targets that are flanked by the NGG PAM. Given decreased single base specificity at the PAM-proximal sites in the CRISPR-Cas9 genome engineering (*Hsu et al., 2013*) and the abundance of CAG repeat carrying genes in the human genome, many of our gRNAs (except gRNAs 1 and 2) are predicted to hybridize with many CAG repeat sequences in the genome, generating increased numbers of predicted off-targets. Thus, we performed experimental validation of (1) predicted off-targets for BE strategies 1 and 2 (described here) and (2) genes that cause polyglutamine disorders. For the experimental validation of predicted off-targets, we analyzed HEK293 DNA samples that were used for MiSeq analysis. Briefly, we focused on predicted off-targets in the protein-coding genes for gRNAs 1 and 2 with two mismatches. One and four potential off-targets were predicted for gRNAs 1 (*MINK1*) and 2 (*PINK1, ZNF704, WBP1L, C20orf112*), respectively. We amplified predicted off-target sites of gRNAs 1 and 2 (35 cycles) using the following primers:

*MINK1*, AGCATGCCTACCTCAAGTCC and CTGGTTTGTCAGCGGGATTC;
*PINK1*, CTGTACCCTGCGCCAGTA and GGATGTTGTCGGATTTCAGGT;
*ZNF704*, GGACGGGTTGGACTGGTC and GGGTCCTGGCACTGACTGTG;
*WBP1L*, CCGACCTCCAACTCCTCCC and GCTGCTCTGTGCCCCCTG; and
*C20orf112*, GATCTCCGTGGGGCTGAG and CCTACTTCCCTCTCCACAGG.

Amplified DNA samples were analyzed by MiSeq sequencing.

## Experimental validation of off-targets in genes causing polyglutamine diseases

Similarly, we amplified genomic regions (35 cycles) containing CAG repeat regions in the genes causing polyglutamine diseases using the following primers:

ATXN1, CCTGCTGAGGTGCTGCTG and CAACATGGGCAGTCTGAGC;
*ATXN2*, CGGGCTTGCGGACATTGG and GTGCGAGCCGGTGTATGG;
*ATXN3*, GAATGGTGAGCAGGCCTTAC and TTCAGACAGCAGCAAAAGCA;
*CACNA1A*, CCTGGGTACCTCCGAGGGC and ACGTGTCCTATTCCCCTGTG;
*ATXN7*, GAAAGAATGTCGGAGCGGG and CTTCAGGACTGGGCAGAGG;
*TBP*, AAGAGCAACAAAGGCAGCAG and AGCTGCCACTGCCTGTTG;
*ATN1*, CCAGTCTCAACACATCACCAT and AGTGGGTGGGGAAATGCTC; and
*AR*, CTCCCGGCGCCAGTTTGCTG and GAACCATCCTCACCCTGCTG.

Sequencing data analysis was focused on calculating the proportion of sequence reads that contain the CAG-to-CAA conversions.

## RNAseq analysis

To determine the molecular consequences of candidate BE strategies, we performed RNAseq analysis. We transfected HEK293 cells with BE4max+EV, BE4max+gRNA 1, or BE4max+gRNA 2 for 72 hr. Subsequently, genomic DNA for MiSeq analysis and cell pellets for RNAseq analysis were generated from replica plates genome-wide RNAseq analysis (Tru-Seq strand-specific large insert RNAseq) was performed by the Broad Institute. Sequence data were processed by STAR aligner (*Dobin et al., 2013*) as part of the Broad Institute's standard RNAseq analysis pipeline. For differential gene expression (DGE) analysis, we used transcripts per million (TPM) data computed by the TPMCalculator (https://github.com/ncbi/TPMCalculator; *Vera Alvarez et al., 2019a*; *Vera Alvarez et al., 2019b*). Expression levels in approximately 19,000 protein-coding genes based on Ensembl (ftp://ftp.ensembl.org/pub/release-75/gtf/homo_sapiens/) were normalized. The DGE analysis was performed by the generalized linear model using a library of 'glm' in R package v3.3.1 (https://www.r-project.org/) after adjustment for two principal components based on RNAseq data, followed by multiple test correction using a false discovery rate method. A multiple test-corrected p-value less than 0.05 was considered statistically significant.

## Generation and validation of HEK293-51 CAG cells carrying an expanded CAG repeat

HD patient-derived iPSC and neurons showed modest conversion efficiencies, making it technically difficult to characterize molecular consequences of CAG-to-CAA conversion strategies. Thus, we generated HEK293 cells carrying an expanded repeat by replacing one of the non-expanded *HTT* CAG repeats with a 51 CAG repeat. Briefly, we cloned a gRNA (CAGAGCGCAGAGAATGCGCG) into the PX459 vector (Addgene# 62988) to express SpCas9 and gRNA for CRISPR-Cas9 targeting at the *HTT* CAG repeat region. The donor template for homologous recombination was generated by PCR amplification of a human DNA sample carrying 51 CAG repeat into the pCR-Blunt II TOPO plasmid (Invitrogen). Subsequently, HEK293 cells were transfected with PX459 and pCR-Blunt II TOPO plasmids by Lipofectamine 3000 (Invitrogen) for 72 hr. Subsequently, cells were treated with G-418 (Gibco) for 21 days, and surviving cells were re-plated onto 100 cm dishes. After 10 days, visible colonies were picked and maintained separately. Single-cell clonal lines were validated by PCR analysis using AccuPrime GC-Rich DNA Polymerase and primer set (ATGAAGGCCTTCGAGTCCC and GGCTGAGGAAGCTGAGGA). The PCR conditions were initial denaturation (95°C, 3 min), 30 cycles of denaturation (95°C, 30 s), annealing (55°C, 30 s), extension (72°C, 40 s), and final extension (72°C, 10 min). The PCR products were resolved on a 1.5% agarose gel containing GelRed (Biotium) and visualized under UV light to distinguish expanded from non-expanded CAG repeats. We also performed RT-PCR and immunoblot analysis to confirm the correct integration of the expanded CAG repeat. Briefly, 1 μg of total RNA from the targeted clonal line was subjected to reverse transcription with SuperScript IV Reverse Transcriptase (Invitrogen) according to the manufacturer's instructions followed by PCR analysis using a primer set (ATGAAGGCCTTCGAGTCCC and GGCTGAGGAAGCTGAGGA). For HTT immunoblot analysis, cells were lysed with RIPA Lysis/Extraction Buffer (Thermo) supplemented with

Halt Protease and Phosphatase Inhibitor Cocktail (Thermo). Whole-cell lysate was then separated on NuPAGE 3–8%, Tris-Acetate gel (Invitrogen) and transferred to a polyvinylidene fluoride membrane. The membrane was blocked with 5% nonfat dry milk in Tris-buffered saline for 1 hr and incubated with primary antibodies for HTT (MAB2166, Sigma-Aldrich) for 12 hr at 4°C. The membrane was washed for 1 hr, and blots were incubated with a peroxidase-conjugated secondary antibody for 1 hr then washed for 1 hr. The bands were visualized by enhanced chemiluminescence (Thermo). Similar to HEK293 cells, HEK293-51CAG cells were treated with BE4max and candidate gRNAs (i.e. gRNA 1 and gRNA 2) to determine the levels of CAG-to-CAA conversion and the total HTT protein levels. For gRNA 1, we determined the levels of in-frame insertion and deletion right after treatment using methods previously described (*Lee et al., 2010*).

### AAV treatment for a candidate BE strategy and CAG repeat instability in mice

For AAV injection experiments, we used split-intein base editor (v5 AAV) (*Levy et al., 2020*). Forward and reverse oligos (CACCGCTGCTGCTGCTGCTGCTGGA and AAACTCCAGCAGCAGCAGCA GCAGC) (IDT) for gRNA 2 were cloned into the BSmBI site of pCbh_v5 AAV-cytosine base editor C-terminal (Addgene, # 137176) and pCbh_v5 AAV-cytosine base editor N-terminal (Addgene, # 137175). Cloned vectors were validated by Sanger sequencing, and subsequently packed into AAV9 serotype by UMass Viral Vector Core. HttQ111 HD knock-in mice (*Wheeler et al., 1999*) were maintained on an FVB/N background (*Lloret et al., 2006*); AAV9 injections were performed in heterozygous HttQ111/+mice at 6–11 weeks. Animal husbandry was performed under controlled temperature and light/dark cycles. After anesthesia was induced using isoflurane, an insulin syringe was inserted into the medial canthus with the bevel of the needle facing down from the eyeball, advanced until the needle tip was at the base of the eye. We injected HD knock-in mice with AAV9 mix (200 µl containing C-terminal and N-terminal split-intein base editor, $1 \times 10^{12}$ viral genome for each) (experimental group) or PBS (200 µl, control group) by retro-orbital injection. Twelve weeks later, liver and tail samples were collected for instability analysis. Briefly, DNA samples were amplified using primer set (6'FAM-ATGAAGGCC TTCGAGTCCCTCAAGTCCTTC and GGCGGCTGAGGAAGCTGAGGA) and analyzed by ABI3730 to determine the sizes of fragments. Quantification of repeat expansion was based on the expansion index method that we developed previously. The expansion index method robustly quantifies the levels of repeat instability by eliminating potential noise in the fragment analysis results based on the relative peak height threshold (*Lee et al., 2011*; *Neugebauer et al., 2023*). To quantify expansion index in control and mice treated with BE, we applied 10% threshold, and expansion index was calculated based on the highest peak in the tail DNA. Since significantly lower delivery/targeting efficiency was expected in the brain (*Levy et al., 2020*), we focused on analyzing liver instability.

### Repeat instability in HD knock-in mice carrying interrupted CAG repeat

To determine the maximal effects of CAG-to-CAA interruption, we analyzed HD knock-in mice carrying interrupted CAG repeat (namely interrupted repeat mice; https://www.jax.org/strain/027418) to HD knock-in mice carrying uninterrupted repeat (namely, pure repeat mice; https://www.jax.org/strain/027417). Repeat in the interrupted repeat mice and pure repeat mice comprises 21 copies of [CAGCAACAGCAACAA] and 105 copies of [CAG], respectively. Both mouse lines were expected to produce huntingtin protein with 105 polyglutamine. Repeat instability in these mice were determined (5 months) by the fragment analysis as described previously (*Lee et al., 2011*).

### Statistical analysis and software

Statistical analysis of RNAseq data was performed using generalized linear regression analysis. Multiple test correction was performed using false discovery rate using R 3.5.3 (*Benjamini et al., 2001*). R 3.5.3 was also used to produce plots.

## Acknowledgements

We thank Drs. Marcy E MacDonald, James F Gusella, and David Liu for helpful discussion. This work was supported by grants from Harvard NeuroDiscovery Center, NIH (NS105709, NS119471, NS049206), and CHDI Foundation. BPK was also supported by an MGH ECOR Howard M Goodman Award and CHDI Foundation.

# Additional information

## Competing interests

Vanessa C Wheeler: V.C.W. was a founding scientific advisory board member with financial interest in Triplet Therapeutics Inc, Her financial interests were reviewed and are managed by Massachusetts General Hospital and Mass General Brigham in accordance with their conflict of interest policies. V.C.W. is a scientific advisory board member of LoQus23 Therapeutics Ltd. and has provided paid consulting services to Acadia Pharmaceuticals Inc, Alnylam Inc, Biogen Inc and Passage Bio. V.C.W. has received research support from Pfizer Inc. Ben Kleinstiver: B.P.K is an inventor on patents and/or patent applications filed by Mass General Brigham that describe genome engineering technologies. B.P.K. is a consultant for EcoR1 capital and is a scientific advisory board member of Acrigen Biosciences, Life Edit Therapeutics, and Prime Medicine. Jong-Min Lee: J-ML consults for GenKOre and serves in the advisory board of GenEdit Inc. The other authors declare that no competing interests exist.

## Funding

| Funder | Grant reference number | Author |
| --- | --- | --- |
| National Institute of Neurological Disorders and Stroke | NS105709 | Jong-Min Lee |
| National Institute of Neurological Disorders and Stroke | NS119471 | Jong-Min Lee |
| National Institute of Neurological Disorders and Stroke | NS049206 | Vanessa C Wheeler |
| CHDI Foundation | | Ben Kleinstiver Jong-Min Lee |

The funders had no role in study design, data collection and interpretation, or the decision to submit the work for publication.

## Author contributions

Doo Eun Choi, Data curation, Formal analysis, Investigation, Writing – original draft; Jun Wan Shin, Jae-Hyun Jang, Investigation, Writing – original draft; Sophia Zeng, Investigation; Eun Pyo Hong, Data curation, Investigation, Writing – original draft; Jacob M Loupe, Vanessa C Wheeler, Hannah E Stutzman, Resources; Ben Kleinstiver, Resources, Writing – original draft; Jong-Min Lee, Conceptualization, Software, Supervision, Funding acquisition, Visualization, Writing – original draft, Project administration, Writing – review and editing

## Author ORCIDs

Eun Pyo Hong ⓘ https://orcid.org/0000-0001-7789-686X
Vanessa C Wheeler ⓘ https://orcid.org/0000-0003-2619-589X
Hannah E Stutzman ⓘ https://orcid.org/0000-0003-0500-437X
Jong-Min Lee ⓘ https://orcid.org/0000-0001-5799-0787

## Ethics

This study analyzed only data from human subjects, not involving new recruitment. The original subject consents and the overall study were approved by the Mass General Brigham IRB and described previously (PMID: 31398342).

Experiments involving mice were approved by the Mass General Brigham Institutional Animal Care and Use Committee (protocol numbers, 2018N000220 and 2020N000135) . Every effort was made to minimize suffering.

Reviewer #1 (Public Review): https://doi.org/10.7554/eLife.89782.2.sa1
Reviewer #2 (Public Review): https://doi.org/10.7554/eLife.89782.2.sa2
Reviewer #3 (Public Review): https://doi.org/10.7554/eLife.89782.2.sa3

Author response https://doi.org/10.7554/eLife.89782.2.sa4

## Additional files

### Supplementary files

• Supplementary file 1. Sequences, expected hybridization sites, and protospacer-adjacent motifs (PAMs) of guide RNAs (gRNAs) used in this study. For each gRNA, sequence, length, hybridization site, and PAM sequence are summarized.

• Supplementary file 2. Types of base conversion by base editing (BE) strategies. HEK293 cells were treated with different combinations of base editors and guide RNAs (gRNAs). Subsequently, genomic DNA samples were analyzed by MiSeq platform to determine sequence changes produced by BE strategies. Base editor-none and gRNA-none represents HEK293 cells without any treatment (n=8). Empty vector represents HEK293 cells treated with plasmids for a base editor and empty vector for gRNA. Data represent means of three independent experiments.

• Supplementary file 3. Sites of CAG-to-CAA conversion in HEK293 cells. HEK293 cells treated with combinations of base editors and guide RNAs (gRNAs) were analyzed by MiSeq. Each sequence read with 16 or 17 CAG repeats were further analyzed to find out sites of conversion. Each data value means the percentage of alleles containing the CAA at a given site relative to all alleles. Base editor-none and gRNA-none represents HEK293 cells without any treatment (n=8). Empty vector represents HEK293 cells treated with a base editor and empty vector for gRNA. Data represent means of three independent experiments.

• Supplementary file 4. The levels of multiple CAG-to-CAA conversion. HEK293 cells were treated with base editors and guide RNA (gRNA) and subsequently analyzed by MiSeq. Each sequence reads of 16 or 17 repeat alleles were analyzed to count the number of CAG-to-CAA conversion in a given repeat. Each data value means the percentage of alleles containing the given number of conversions relative to all alleles. Base editor-none and gRNA-none represents HEK293 cells without any treatment (n=8). Empty vector represents HEK293 cells treated with plasmid for a base editor and empty vector for gRNA. Data represent means of three independent experiments.

• Supplementary file 5. Off-target prediction. The number of predicted off-targets are summarized. MM represents mismatch. Numbers in parentheses shows predicted off-targets on the protein-coding genes.

• Supplementary file 6. Experimental validation of predicted off-targets. HEK293 cells that were treated with base editing (BE) strategies 1 or 2. Subsequently, we performed MiSeq analysis focusing on predicted off-targets on protein-coding genes. Locations of off-targets are based on hg19. MM represents the number of mismatches. MiSeq analysis was performed for HTT as on-target. Numbers represent the percentages of sequence reads containing CAG-to-CAA conversion.

• Supplementary file 7. Other polyglutamine disease genes. The names of polyglutamine diseases, respective genes, and RefSeq IDs are shown.

• Supplementary file 8. Off-target conversions in other polyglutamine disease genes. Representative HEK293 cell MiSeq data were analyzed to determine the levels of CAG-to-CAA conversion in other polyglutamine disease genes. Each number represents the percentage of sequence reads that contained CAG-to-CAA conversion relative to all sequence reads.

• Supplementary file 9. CAG-to-CAA conversion in Huntington's disease (HD) patient-derived cells. Cells were transfected with plasmids for BE4max and guide RNAs (gRNAs). Subsequently, genomic DNA samples were subjected to MiSeq analysis. Each number represents the percentage of sequence reads containing CAA at specific site relative to all quality control (QC)-passed sequence reads.

• MDAR checklist

### Data availability

RNAseq data of control and targeted iPSC clones have been deposited in Dryad (https://doi.org/10.5061/dryad.k3j9kd5cb).

The following dataset was generated:

| Author(s) | Year | Dataset title | Dataset URL | Database and Identifier |
|---|---|---|---|---|
| Lee JM | 2024 | Base editing strategies to convert CAG to CAA diminish the disease-causing mutation in Huntington's disease | http://doi.org/10.5061/dryad.k3j9kd5cb | Dryad Digital Repository, 10.5061/dryad.k3j9kd5cb |

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
