## [Editor Report · eLife Assessment]

This proof-of-concept study focuses on an A->G DNA base editing strategy that converts CAG repeats to CAA repeats in the human HTT gene, which causes Huntington's disease (HD). These studies are conducted in human HEK293 cells engineered with a 51 CAG canonical repeat and in HD knock-in mice harboring 105+ CAG repeats. The findings of this study are **valuable** for the HD field, applying state-of-the-art techniques; however, the key experiments have yet to be performed in neuronal systems or brains of these mice: actual disease-rectifying effects relevant to patients have yet to observed, leaving the work **incomplete**.

---

## [Referee Report · Reviewer #1 (Public Review)]

Summary:

In the paper by Choi et al., the authors aimed to develop base editing strategies to convert CAG repeats to CAA repeats in the huntingtin gene (HTT), which causes Huntington's disease (HD). They hypothesized that this conversion would delay disease onset by shortening the uninterrupted CAG repeat. Using HEK-293T cells as a model, the researchers employed cytosine base editors and guide RNAs (gRNAs) to efficiently convert CAG to CAA at various sites within the CAG repeat. No significant indels, off-target edits, transcriptome alterations, or changes in HTT protein levels were detected. Interestingly, somatic CAG repeat expansion was completely abolished in HD knock-in mice carrying CAA-interrupted repeats.

Strengths:

This study represents the first proof-of-concept exploration of the cytosine base editing technique as a potential treatment for HD and other repeat expansion disorders with similar mechanisms.

Weaknesses:

Given that HD is a neurodegenerative disorder, it is crucial to determine the efficiency of the base editing strategies tested in this manuscript and their feasibility in relevant cells affected by HD and the brain, which needed to be improved in this manuscript.

---

## [Referee Report · Reviewer #2 (Public Review)]

Summary:

In a proof-of-concept study with the aspiration of developing a treatment to delay HD onset, Choi et al. design and test an A>G DNA base editing strategy to exploit the recently established inverse relationship between the number of uninterrupted CAG repeats in polyglutamine repeat expansions and the age-of-onset of Huntington's Disease (HD). Most of the study is devoted to optimizing a base editing strategy typified by BE4max and gRNA2. The base editing is performed in human HEK293 cells engineered with a 51 CAG canonical repeat and in HD knock-in mice harboring 105+ CAG repeats.

Weaknesses:

Genotypic data on DNA editing are not portrayed in a clear manner consistent with the study's goal, namely reducing the number of uninterrupted CAG repeats by a clinically relevant amount according to the authors' least square approximated mean age-at-onset. No phenotypic data are presented to show that editing performed in either model would lead to reduced hallmarks of HD onset.

More evidence is needed to support the central claims and therapeutic potential needs to be more adequate.

---

## [Referee Report · Reviewer #3 (Public Review)]

Summary:

In human patients with Huntington's disease (HD), caused by a CAG repeat expansion mutation, the number of uninterrupted CAG repeats at the genomic level influences age-at-onset of clinical signs independent of the number of polyglutamine repeats at the protein level. In most patients, the CAG repeat terminates with a CAA-CAG doublet. However, CAG repeat variants exist that either do not have that doublet or have two doublets. These variants consequently differ in their number of uninterrupted CAG repeats, while the number of glutamine repeats is the same as both CAA and CAG codes for glutamine. The authors first confirm that a shorter uninterrupted CAG repeat number in human HD patients is associated with developing the first clinical signs of HD later. They predict that introducing a further CAA-CAG doublet will result in years of delay of clinical onset. Based on this observation, the authors tested the hypothesis that turning CAG to CAA within a CAG repeat sequence using base editing techniques will benefit HD biology. They show that, indeed, in HD cell models (HEK293 cells expressing 16/17 CAG repeats; a single human stem cell line carrying a CAG repeat expansion in the fully penetrant range with 42 CAG repeats), their base editing strategies do induce the desired CAG-CAA conversion. The efficiency of conversion differed depending on the strategy used. In stem cells, delivery posed a problem, so to test allele specificity, the authors then used a HEK 293 cell line with 51 CAG repeats on the expanded allele. Conversion occurred in both alleles with huntingtin protein and mRNA levels; transcriptomics data was unchanged. In knock-in mice carrying 110 CAG repeats, however, base editing did not work as well for different, mainly technical, reasons.

Strengths:

The authors use state-of-the-art methods and carefully and thoroughly designed experiments. The data support the conclusions drawn. This work is a very valuable translation from the insight gained from large GWAS studies into HD pathogenesis. It rightly emphasises the potential this has as a causal treatment in HD, while the authors also acknowledge important limitations.

Weaknesses:

They could dedicate a little more to discussing several of the mentioned challenges. The reader will better understand where base editing is in HD currently and what needs to be done before it can be considered a treatment option. For instance,

-It is important to clarify what can be gained by examining again the relationship between uninterrupted CAG repeat length and age-at-onset. Could the authors clarify why they do this and what it adds to their already published GWAS findings? What is the n of datasets?

-What do they think an ideal conversion rate would be, and how that could be achieved?

-Is there a dose-effect relationship for base editing, and would it be realistic to achieve the ideal conversion rate in target cells, given the difficulties described by the authors in differentiated neurons from stem cells?

- The liver is a good tool for in-vivo experiments examining repeat instability in mouse models. However, the authors could comment on why they did not examine the brain.

- Is there a limit to judging the effects of base editing on somatic instability with longer repeats, given the difficulties in measuring long CAG repeat expansions?

- Given the methodological challenges for assessing HTT fragments, are there other ways to measure the downstream effects of base editing rather than extrapolate what it will likely be?

- Sequencing errors could mask low-level, but biologically still relevant, off-target effects (such as gRNA-dependent and gRNA-independent DNA, Off-targets, RNA off-targets, bystander editing). How likely is that?

- How worried are the authors about immune responses following base editing? How could this be assessed?

---

## [Author Response]

**Public Reviews:**

**Reviewer #1 (Public Review):**
Summary:In the paper by Choi et al., the authors aimed to develop base editing strategies to convert CAG repeats to CAA repeats in the huntingtin gene (HTT), which causes Huntington's disease (HD). They hypothesized that this conversion would delay disease onset by shortening the uninterrupted CAG repeat. Using HEK-293T cells as a model, the researchers employed cytosine base editors and guide RNAs (gRNAs) to efficiently convert CAG to CAA at various sites within the CAG repeat. No significant indels, off-target edits, transcriptome alterations, or changes in HTT protein levels were detected. Interestingly, somatic CAG repeat expansion was completely abolished in HD knock-in mice carrying CAA-interrupted repeats.

Correction of factual errors

We analyzed HEK293 cells, not "HEK-293T".

Strengths:This study represents the first proof-of-concept exploration of the cytosine base editing technique as a potential treatment for HD and other repeat expansion disorders with similar mechanisms.Weaknesses:Given that HD is a neurodegenerative disorder, it is crucial to determine the efficiency of the base editing strategies tested in this manuscript and their feasibility in relevant cells affected by HD and the brain, which needed to be improved in this manuscript.

We appreciate the reviewer's constructive recommendations. Our genetic investigation focused on understanding observations in HD patients to develop genetic-based treatment strategies and test their feasibility. We agree with the reviewer regarding the importance of data from relevant cell types. Unfortunately, the levels of CAG-to-CAA conversion in the patient-derived neurons were modest, as described in our manuscript (approximately 2%). In addition, AAV did not produce detectable conversions in the brain of HD knock-in mice (data not shown), which was somewhat expected from the literature (PMID: 31937940). We believe some technical hurdles can be overcome by developing efficient delivery methods. Nonetheless, it will be an important follow-up study to perform preclinical studies employing optimized base editing strategies and efficient brain delivery methods to fully demonstrate the therapeutic potential of BE strategies.

**Reviewer #2 (Public Review):**
Summary:In a proof-of-concept study with the aspiration of developing a treatment to delay HD onset, Choi et al. design and test an A>G DNA base editing strategy to exploit the recently established inverse relationship between the number of uninterrupted CAG repeats in polyglutamine repeat expansions and the age-of-onset of Huntington's Disease (HD). Most of the study is devoted to optimizing a base editing strategy typified by BE4max and gRNA2. The base editing is performed in human HEK293 cells engineered with a 51 CAG canonical repeat and in HD knock-in mice harboring 105+ CAG repeats.

Correction of factual errors

We tested base editing strategies aimed at C > T conversion, not A > G DNA base editing. In addition to HEK293 and knock-in mice, we tested base editing strategies in patient-derived iPSC and neurons.

Weaknesses:Genotypic data on DNA editing are not portrayed in a clear manner consistent with the study's goal, namely reducing the number of uninterrupted CAG repeats by a clinically relevant amount according to the authors' least square approximated mean age-at-onset. No phenotypic data are presented to show that editing performed in either model would lead to reduced hallmarks of HD onset.More evidence is needed to support the central claims and therapeutic potential needs to be more adequate.

Our strategies for converting CAG to CAA in model systems resulted in quantitative DNA modification in a population of cells. Consequently, individual cells may carry different genotypes, some harboring CAA and others CAG at the same genomic location. Therefore, using a standard genotype format for DNA to present base editing outcomes may not be ideal. Instead, we presented the resulting genotype data in a quantitative fashion to provide the percentage of conversion at each site. This approach allows for an intuitive interpretation of both the extent of repeat length reduction and the proportion of such modifications.

Currently, genetically precise HD mouse models with robust motor and behavioral phenotypes are unavailable. While some HD mouse models, such as the BAC and YAC models, feature pronounced behavioral phenotypes, they consist of interrupted CAG repeat sequences, making them unsuitable for base conversion studies due to their inherently short uninterrupted repeats. Although genetically precise HD knockin mouse models exist, they do not manifest motor symptom-like phenotypes. Given that CAG repeat expansion is the primary driver of the disease and knock-in mice recapitulate such phenomenon, our genetic investigation focused on assessing the effects of base conversion on CAG repeat instability in knock-in mice. However, as emphasized by the reviewer, subsequent preclinical studies to evaluate the therapeutic efficacy of CAG-to-CAA conversion strategies using mouse models harboring uninterrupted adult-onset CAG repeats and robust HD-like phenotypes remain crucial.

**Reviewer #3 (Public Review):**
Summary:In human patients with Huntington's disease (HD), caused by a CAG repeat expansion mutation, the number of uninterrupted CAG repeats at the genomic level influences age-at-onset of clinical signs independent of the number of polyglutamine repeats at the protein level. In most patients, the CAG repeat terminates with a CAACAG doublet. However, CAG repeat variants exist that either do not have that doublet or have two doublets. These variants consequently differ in their number of uninterrupted CAG repeats, while the number of glutamine repeats is the same as both CAA and CAG codes for glutamine. The authors first confirm that a shorter uninterrupted CAG repeat number in human HD patients is associated with developing the first clinical signs of HD later. They predict that introducing a further CAA-CAG doublet will result in years of delay of clinical onset. Based on this observation, the authors tested the hypothesis that turning CAG to CAA within a CAG repeat sequence using base editing techniques will benefit HD biology. They show that, indeed, in HD cell models (HEK293 cells expressing 16/17 CAG repeats; a single human stem cell line carrying a CAG repeat expansion in the fully penetrant range with 42 CAG repeats), their base editing strategies do induce the desired CAG-CAA conversion. The efficiency of conversion differed depending on the strategy used. In stem cells, delivery posed a problem, so to test allele specificity, the authors then used a HEK 293 cell line with 51 CAG repeats on the expanded allele. Conversion occurred in both alleles with huntingtin protein and mRNA levels; transcriptomics data was unchanged. In knock-in mice carrying 110 CAG repeats, however, base editing did not work as well for different, mainly technical, reasons.

Correction of factual errors

"HD cell models HEK293 cells expressing 16/17 CAG repeats" is an incorrect description. It should be "HD cell models HEK293 cells expressing 51/17 CAG repeats".

Strengths:The authors use state-of-the-art methods and carefully and thoroughly designed experiments. The data support the conclusions drawn. This work is a very valuable translation from the insight gained from large GWAS studies into HD pathogenesis. It rightly emphasises the potential this has as a causal treatment in HD, while the authors also acknowledge important limitations.Weaknesses:They could dedicate a little more to discussing several of the mentioned challenges. The reader will better understand where base editing is in HD currently and what needs to be done before it can be considered a treatment option. For instance,- It is important to clarify what can be gained by examining again the relationship between uninterrupted CAG repeat length and age-at-onset. Could the authors clarify why they do this and what it adds to their already published GWAS findings? What is the n of datasets?

Published HD GWAS (PMID: 31398342) compared the onset age of duplicated interruption and loss of interruption to that of canonical repeats to determine whether uninterrupted CAG repeat or polyglutamine determines age at onset. However, GWAS findings did not quantify the magnitude of the unexplained remaining variance in age at onset in duplicated interruption and loss of interruption. Our study further investigated to gain insights into the amount of additional impact of duplicated interruption to estimate the maximum clinical benefits of base editing strategies for CAG-to-CAA conversion. Since the purpose of this genetic analysis is described in the result section already, we added the following sentence in the introduction section to bring up what is unknown.

"Still, age at onset of loss of interruption and duplicated interruption was not fully accounted for by uninterrupted CAG repeat, suggesting additional effects of non-canonical repeats."

We added sample size for the least square approximation analysis in the text and corresponding figure legend. Sample sizes for molecular and animal experiments can be found in the corresponding figure legend.

- What do they think an ideal conversion rate would be, and how that could be achieved?

It is a very important question. However, speculating the ideal conversion levels is out of the scope of this genetic investigation. A series of preclinical studies using relevant models may generate data that may shed light on the conversion rate levels that are required to produce meaningful clinical benefits. In the discussion section, we added the following sentence.

"Currently, the ideal levels of CAG-to-CAA conversion that produce significant clinical benefits are unknown. A series of preclinical studies using relevant model systems may generate data that may shed light on the optimal conversion rate levels that are required to produce significant clinical benefits."

- Is there a dose-effect relationship for base editing, and would it be realistic to achieve the ideal conversion rate in target cells, given the difficulties described by the authors in differentiated neurons from stem cells?

We observed a clear dose-response relationship between the amount of BE reagents and the levels of conversion in non-neuronal cells. Unfortunately, the conversion rate was low in neuronal cells, potentially due to limited delivery, as speculated in the result section. As described in the discussion sections, we predict that efficient delivery methods will be crucial to produce significant CAG-to-CAA conversion to achieve therapeutic benefits.

- The liver is a good tool for in-vivo experiments examining repeat instability in mouse models. However, the authors could comment on why they did not examine the brain.

We focused on liver instability because of (1) the expectation that delivery/targeting efficiency is significantly lower in the brain (PMID: 31937940) and (2) shared underlying mechanisms between the brain and liver (described in the result section). The following sentence was added in the method section to provide a rationale for liver analysis.

"Since significantly lower delivery/targeting efficiency was expected in the brain 34, we focused on analyzing liver instability."

- Is there a limit to judging the effects of base editing on somatic instability with longer repeats, given the difficulties in measuring long CAG repeat expansions?

Determining the levels of base conversion using sequencing technologies gets harder as repeats become longer. Fragment analysis can overcome such technical difficulty if conversion efficiency is high. As pointed out, the repeat expansion measure is also challenging because amplification is biased toward shorter alleles. However, if repeat sizes are relatively similar, the levels of repeat expansion as a function of base conversion can be determined relatively precisely without a significant bias by a standard fragment analysis approach.

- Given the methodological challenges for assessing HTT fragments, are there other ways to measure the downstream effects of base editing rather than extrapolate what it will likely be?

Our CAG-to-CAA conversion strategies are not expected to directly generate fragments of huntingtin DNA, RNA, or protein. In contrast, immediate downstream effects of CAG-to-CAA conversion include sequence changes (DNA and RNA) and alteration of repeat instability, which are presented in the manuscript. If repeat instability is associated with HTT exon 1A fragment, base conversion strategies may indirectly alter the levels of such putative toxic species, which remains to be determined.

- Sequencing errors could mask low-level, but biologically still relevant, off-target effects (such as gRNAdependent and gRNA-independent DNA, Off-targets, RNA off-targets, bystander editing). How likely is that?

We agree with the reviewer that increased editing efficiency is expected to increase the levels of off-target editing. However, the field is actively developing base editors with minimal off-target effect (PMID: 35941130), which will increase the safety aspects of this technology for clinical use. We added the following sentence. "In addition, developing base editors with high level on-target gene specificity and minimal off-target effects is a critical aspect to address 100."

- How worried are the authors about immune responses following base editing? How could this be assessed?

We added the following sentence in the discussion section as the reviewer raised an important safety issue.

"Thorough assessments of immune responses against base editing strategies (e.g., development of antibody, B cell, and T cell-specific immune responses) and subsequent modification (e.g., immunosilencing) 101 will be critical to address immune response-associated safety issues of BE strategies."

**Recommendations for the authors:**

**Reviewer #1 (Recommendations For The Authors):**
The following points could be considered to improve the overall quality of the manuscript:(1) The authors mentioned that the reason for checking repeat instability in the nonneuronal cells was due to the availability of specific types of AAV; there are other subtypes of AAVs available to infect neurons and iPSCs.

Our pilot experiments testing several AAV serotypes in patient-derived iPSC and HD knock-in mice showed that only AAV9 converted CAG to CAA at detectable levels in the liver, not in the brain or neurons. We also speculate that difficulties in targeting the CAG repeat region due to GC-rich sequence contributed to low conversion efficiency. Therefore, subsequent optimization of base editor and delivery may improve BE strategies for HD, permitting robust conversion at the challenging locus.

(2) Despite its bold nature, minimal data in the manuscript demonstrate that this gene editing strategy is disease-modifying.

Resources required to demonstrate the therapeutic benefits of CAG-to-CAA conversion strategies are not fully available. Especially, relevant HD mouse models that carry uninterrupted adult onset CAG repeat and that permit measuring the levels of disease-modifying are lacking, as described in our response to the second reviewer. Given that CAG repeat expansion is the primary driver of the disease, this genetic investigation focused on determining the impacts of base editing strategies on CAG repeat expansion. Still, as indicated by the reviewer, follow-up preclinical studies to evaluate the levels of disease-modifying of CAG-to-CAA conversion strategies using relevant mouse models represent important next steps.

(3) Off-target analysis at the DNA level was limited to "predicted" off-target sites. What about possible translocations that can result from co-nicking on different chromosomes, as a large number of potential targets exist?

Among gRNAs we tested, we focused on gRNAs 1 and 2, which predicted small numbers of off-target. Therefore, our off-target analysis at the DNA level was focused on validating those predicted off-targets. As pointed out, thoroughly evaluating off-target effects will be necessary when candidate BE strategies take the next steps for therapeutic development.

Genomic translocation caused by double-strand breaks can produce negative consequences, such as cancer. Importantly, although paired nicks efficiently induced translocations, translocations were not detected when a single nick was introduced on each chromosome (PMID: 25201414). Therefore, it is predicted that BE strategies using nickase confers little risk of translocation.

(4) For in vivo work, somatic repeat expansion was analyzed only in peripheral tissue samples. Since the main affected cellular population in HD is the brain, the outcome of this treatment on a disease-relevant organ still needs to be determined.

Challenges in delivery to the brain made us determine instability in the liver since many mechanistic components of somatic CAG repeat instability are shared between the liver and striatum, as rationalized in the manuscript. However, we agree with the reviewer regarding the importance of determining the effects of base conversion on brain instability. We added the following sentence in the method section to provide a rationale. "Since significantly lower delivery/targeting efficiency was expected in brain 34, we focused on analyzing liver instability."

**Reviewer #2 (Recommendations For The Authors):**
Throughout the manuscript, the authors apologize for techniques that do not work when workarounds seem readily apparent to an expert in the field. In its current form, the manuscript reads verbose, speculative, apologetic, and preliminary.

Drug development programs that are supported by human genetics data show increased success rates in clinical trials (PMID: 26121088, 31827124, 31830040). This is why this genetic study focused on (1) investigating observations in HD subjects and (2) subsequently developing treatment strategies that are supported by patient genetics. As the first illustration of base editing in HD, the main scope of our manuscript is to justify the genetic rationale of CAG-to-CAA conversion and demonstrate the feasibility of therapeutic strategies rooted in patient genetics. As our study was not aimed at entirely demonstrating the clinical benefits of base editing strategies in HD, some of our data were based on tools and approaches that were not fully optimal. We agree with the reviewer that it will be an important next step to employ optimized approaches to evaluate the efficacy of base editing strategies in model systems. Nevertheless, our novel base conversion strategies derived from HD patient genetics represent a significant advancement as they may contribute to developing effective treatments for this devastating disorder.

**Reviewer#3 (Recommendations For The Authors):**
It would make for an easier read if abbreviations were kept to a minimum.

As recommended, we decreased the use of abbreviations. The following has been spelled out throughout the manuscript: CR (canonical repeat), LI (loss of interruption), DI (duplicated interruption), and CBE (cytosine base editor). Other abbreviations with infrequent usage (e.g., ABE, SS, QC) were also spelled out in the text.